# Single-molecule reconstruction of eukaryotic factor-dependent transcription termination

Ying Xiong[1,2,3,11], Weijing Han [2,11], Chunhua Xu[1,11], Jing Shi[4,5,11], Lisha Wang[2], Taoli Jin[2], Qi Jia[2], Ying Lu [1], Shuxin Hu[1], Shuo-Xing Dou [1,3], Wei Lin [4,6,7,8] ✉, Terence R. Strick [9,10] ✉, Shuang Wang [1,2] ✉ & Ming Li [1,2]

Factor-dependent termination uses molecular motors to remodel transcription machineries, but the associated mechanisms, especially in eukaryotes, are poorly understood. Here we use single-molecule fluorescence assays to characterize in real time the composition and the catalytic states of *Saccharomyces cerevisiae* transcription termination complexes remodeled by Sen1 helicase. We confirm that Sen1 takes the RNA transcript as its substrate and translocates along it by hydrolyzing multiple ATPs to form an intermediate with a stalled RNA polymerase II (Pol II) transcription elongation complex (TEC). We show that this intermediate dissociates upon hydrolysis of a single ATP leading to dissociation of Sen1 and RNA, after which Sen1 remains bound to the RNA. We find that Pol II ends up in a variety of states: dissociating from the DNA substrate, which is facilitated by transcription bubble rewinding, being retained to the DNA substrate, or diffusing along the DNA substrate. Our results provide a complete quantitative framework for understanding the mechanism of Sen1-dependent transcription termination in eukaryotes.

Transcription termination defines borders between genes and thereby contributes to regulating RNA production levels from bacteria to human. Unsuccessful termination can lead to the accumulation of aberrant RNA transcripts and unexpected expression of downstream genes. To avoid such interferences, *S. cerevisiae* has evolved two main pathways to terminate transcription: the poly(A)-dependent pathway for messenger RNAs relying on Rat1/Xrn2 exonuclease[1,2], and the Nrd1-Nab3-Sen1 (NNS) complex-dependent pathway for non-coding RNAs[3,4] which are pervasively transcribed in eukaryotes[4–9]. Both pathways require enzymatic actions on RNA polymerase II

(Pol II) in response to termination signals, i.e., factor-dependent termination.

In the NNS pathway, Nrd1 and Nab3 specifically bind to conserved motifs enriched in non-coding RNAs, respectively, GUAA/G and UCUU[10]. Nrd1 interacts with Pol II's carboxy-terminal domain (CTD), which may thus be involved in recruiting the NNS complex and driving NNS-dependent termination[11,12]. Because Sen1 is less abundant in the cell than Nrd1 and Nab3, Sen1 recruitment is possibly a rate-limiting step of NNS-dependent termination[13,14]. In vitro characterizations show that Sen1 or even its helicase domain alone (residues 1095–1904) can

[1]Beijing National Laboratory for Condensed Matter Physics, Institute of Physics, Chinese Academy of Sciences, Beijing, China. [2]Songshan Lake Materials Laboratory, Dongguan, Guangdong, China. [3]School of Physics, University of Chinese Academy of Sciences, Beijing, China. [4]Department of Pathogen Biology, School of Medicine & Holistic Integrative Medicine, Nanjing University of Chinese Medicine, Nanjing, China. [5]Department of Pathology of Sir Run Run Shaw Hospital, Zhejiang University School of Medicine, Hangzhou, China. [6]Jiangsu Collaborative Innovation Center of Chinese Medicinal Resources Industrialization, Nanjing, China. [7]State Key Laboratory of Natural Medicines, China Pharmaceutical University, Nanjing, China. [8]State Key Laboratory of Microbial Resources, Institute of Microbiology, Chinese Academy of Sciences, Beijing, China. [9]Institut de Biologie de l'Ecole Normale Supérieure, PSL Université, INSERM, CNRS, Paris, France. [10]Equipe Labellisée de la Ligue Nationale Contre le Cancer, Paris, France. [11]These authors contributed equally: Ying Xiong, Weijing Han, Chunhua Xu, Jing Shi. ✉e-mail: weilin@njucm.edu.cn; strick@bio.ens.psl.eu; shuangwang@iphy.ac.cn

terminate Pol II transcription in the absence of Nrd1 and Nab3 but requiring ATP hydrolysis and the nascent RNA transcript[15–17]. These results suggest a rho-like model wherein Sen1 possibly acts like a motor translocating along nascent RNA to chase off a Pol II TEC[1,2,15,18,19]. Further studies show that the termination process involves a Sen1-Pol II TEC intermediate conceptually similar to that formed during ρ-mediated transcription termination in *E. coli*[17,20,21]. However, how efficiently Sen1 uses ATP hydrolysis to remodel Pol II TECs and, importantly, the fate of each component (Sen1, Pol II, RNA, and DNA) during Sen1-dependent termination remains elusive.

Single-molecule assays have been extensively applied to the study of both prokaryotic and eukaryotic transcription[22–28]. Because of the low efficiency of eukaryotic transcription initiation[29], promoter-independent transcription assays have been developed to study the mechanisms of eukaryotic transcription elongation and termination with high efficiency[17,22,30]. By using these assays, we have previously characterized the kinetics of the Sen1-dependent termination process and noticed the possible formation of a Sen1-Pol II TEC intermediate[17]. To gain further insights into the mechanism of Sen1-dependent termination, here we use single-molecule fluorescence assays to monitor in real time arrival and departure of the different components of the reaction. We thus not only confirm the presence of a Sen1-Pol II TEC intermediate but also determine the fate of each component of the intermediate, i.e., Sen1, Pol II, RNA transcript, and DNA, throughout the termination process. In addition, we characterize the kinetics of formation and resolution of the Sen1-Pol II TEC intermediate and reveal the effects of ATP usage by Sen1 as well as the rewinding of Pol II transcription bubble on Sen1-dependent termination. Our results provide a complete, quantitative view of Sen1-dependent transcription termination at single-molecule resolution and mechanistic insight into factor-dependent termination for eukaryotic transcription.

## Results

### Sen1 efficiently releases RNA transcript

To explore the mechanism of Sen1-dependent Pol II termination, we use single-molecule fluorescence colocalization assays[31] to characterize Sen1 action on Pol II TEC. Promoter-dependent Pol II initiation is bypassed by sequentially assembling RNA, template DNA, Pol II, and non-template DNA to form Pol II TECs as previously reported (Fig. 1a)[16,22,30]. A G-less cassette which Pol II can transcribe when only ATP, UTP, and CTP are present is positioned in the direction of Pol II transcription, and is followed by a G-stretch which stalls Pol II again if GTP is withheld. Such stalled Pol II TECs can be preferentially terminated by Sen1 helicase as previously report[15]. Different transcription units can be implemented by altering the sequence of the G-less cassette, allowing us to generate different RNA transcript lengths (e.g., 14, 28, and 58 bases). Labeling strategies to make fluorescent or biotinylate each component of Pol II TECs (RNA, DNA, and Pol II) are implemented depending on experimental purposes (Supplementary Data 1). The prepared Pol II TECs possess transcription activity and can be terminated by Sen1 as characterized via an in vitro transcription assay (Supplementary Fig. 1).

For the purposes of single-molecule fluorescence analysis, Pol II TECs assembled using RNA-Cy3 and DNA-Cy5 (at +3 position of the non-template strand) are tethered to a PEG surface via a biotin moiety on Pol II (bioPol II/RNA-Cy3/DNA-Cy5(+3), Supplementary Data 1) and imaged using a homemade TIRF microscope. We can typically image hundreds of well-colocalized pairs of green and red fluorophores with initial Förster resonance energy transfer (FRET) values of ~0.6 under green laser excitation, as well as stable Cy5 signals under alternative red laser excitation (alternating-laser excitation modulation, ALEX, Supplementary Fig. 2a). Addition of 1.1 mM ATP, 100 μM each of UTP and CTP, and 2 nM TFIIS, an anti-backtracking factor, causes changes in Cy3/Cy5 FRET values under green laser excitation as well as protein-induced fluorescence

enhancement (PIFE) of Cy5 under alternating red laser excitation (501 out of 841 trajectories, Supplementary Fig. 2a, b)[32–34]. The FRET changes, increasing initially from ~0.6 to ~0.8 and then decreasing to a final value of ~0.2, reflect the action of Pol II TECs, in agreement with the results from in vitro transcription assays (Supplementary Fig. 1). The time required for the FRET value to change from ~0.6 to ~0.2 is analyzed and seen to be well-described by a single-Gaussian function yielding a peak at $2.9 \pm 0.2$ s (standard error of mean, SEM, Fig. 1c and Supplementary Notes). Taking into account the 28-bp length of the transcription unit, the Pol II transcription rate can be estimated as ~9.7 bp s$^{-1}$, consistent with expectations[17,22]. We infer that the FRET changes indeed correspond to Pol II transcription to the stalling site, and that the durations of FRET changes essentially reflect the length of RNA transcript generated by Pol II transcription. Although a majority of Pol II TECs have final FRET values stabilizing at ~0.2 to the end of the record (180 s, 389 out of 501 trajectories) which suggests stably stalled Pol II TECs, 112 out of 501 Pol II TECs show FRET fluctuations reflecting a backward and forward translocation behavior of Pol II (Supplementary Fig. 2b). Positioning Cy5 from +3 to −16 on the non-template DNA strand (bioPol II/RNA-Cy3/DNA-Cy5(-16), Supplementary Data 1) and supplementing with three nucleotides and TFIIS results in a FRET decrease directly from ~0.5 to ~0.1 which again indicates Pol II transcription (Supplementary Fig. 2c). The appearance of Cy5 PIFE is independent of Cy3 presence (Supplementary Fig. 2d), but its level calculated as the ratio of the mean intensity of Cy5 fluorescence with PIFE over that without PIFE varies slightly when Cy5 is positioned differently (Fig. 1d and Supplementary Fig. 2).

Addition of 10 nM Sen1 helicase domain (HD, residues 1095–1904), 1.1 mM ATP, 100 μM each of UTP and CTP, and 2 nM TFIIS causes the same changes in FRET and Cy5 PIFE as above but faster disappearance of Cy3 fluorescence (~16 s, Fig. 1b, e) compared to the mean time of Cy3 photobleaching (~237 s, Supplementary Fig. 3)[35]. Because of the Cy5 PIFE during Pol II elongation state (gray region), the intensity of Cy5 fluorescence under green laser excitation is possibly overestimated by a factor of 1.83 at 10 nM Sen1 HD concentration. The FRET values could be corrected by using Eq. (1) showing slightly reduced values (light blue in Fig. 1b). The disappearance of Cy3 fluorescence thus reflects RNA release as a result of Sen1 HD terminating Pol II transcription (Fig. 1b and Supplementary Fig. 4). Only very rare termination events are observed before the final FRET value of ~0.2 is reached, in agreement with the requirement of a sufficiently long RNA for Sen1-dependent termination[15]. As the final FRET value of ~0.2 reflects a stalled Pol II TEC, the time for which FRET remains at ~0.2 before Cy3 disappearance reflects the time needed for Sen1 HD to bind to and then catalyze termination of the stalled Pol II TEC. This duration is composed of two sequential steps: Sen1 HD binding and catalysis, and is distributed as per a single exponential decay, suggesting that the binding of Sen1 HD to stalled Pol II TECs is the rate-limiting step and thus the binding rate can be estimated by fitting the time distributions to a single exponential function (Fig. 1d)[36]. By progressively decreasing Sen1 HD concentration from 10 to 3.5 nM, we observe a gradual reduction in the binding rate of Sen1 HD, which is estimated by globally fitting to a single exponential function (Eq. (2)) yielding a binding rate constant of $k_+ = (5.5 \pm 0.2) \times 10^6$ M$^{-1}$ s$^{-1}$ (see "Methods", Supplementary Notes, and Supplementary Fig. 4a).

We also observe termination on Pol II TECs tethered via a biotinylated template DNA to the PEG surface (Pol II/RNA-Cy3/bioDNA-Cy5(+3), Supplementary Fig. 5). Sen1-dependent termination observed on Pol II TECs tethered to the PEG surface either through biotinylated Pol II or biotinylated DNA reflects a complete separation of RNA transcript from Pol II and DNA, which will be discussed in detail below. We note that the presence of Sen1 HD has a negligible effect on the Pol II elongation rate, as elongation kinetics remain statistically identical at various Sen1 HD concentrations (Fig. 1c).

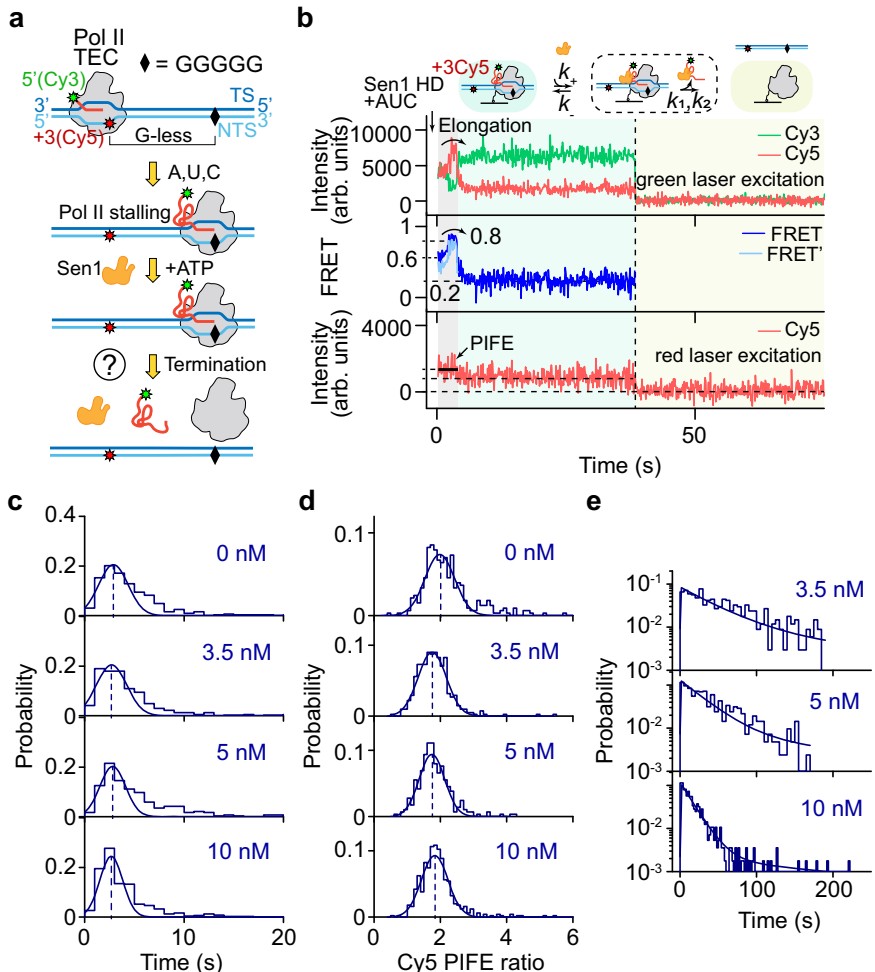

**Fig. 1 | Single-molecule fluorescence characterization of Sen1-mediated Pol II transcription termination. a** Schematic of Pol II transcription initiated in the presence of ATP, UTP, and CTP, stalling, and extrinsic termination mediated by Sen1. **b** Typical trajectories showing (i) Pol II transcription elongation (light-gray zone) as reflected by the changes in Cy3/Cy5 fluorescence and in the corresponding FRET signal (~0.6 → ~0.8 → ~0.2, blue, for FRET and ~0.5 → ~0.7 → ~0.2, light blue, for corrected FRET') in the upper two panels (green laser excitation) and Cy5 PIFE in the lower panel (red laser excitation), followed by (ii) Pol II stalling at the G-stretch (green zone) as reflected by stable Cy3/Cy5 fluorescence and FRET until (iii) Sen1 HD-mediated transcription termination as reflected by the abrupt disappearance of Cy3 fluorescence (vertical dash line). Simultaneous dissociation of DNA-Cy5 is shown in the lower panel. **c** Duration histograms of Pol II elongation (light-gray zone) are each fit to a single-Gaussian function, yielding peaks at 2.9 ± 0.2 s (SEM,

$N = 328$), 2.7 ± 0.3 s (SEM, $N = 412$), 2.8 ± 0.2 s (SEM, $N = 288$) and 2.7 ± 0.1 s (SEM, $N = 498$) for 0, 3.5, 5, and 10 nM Sen1 HD, respectively. The fraction of Pol II restarting elongation is 328 out of 668 in the absence of Sen1 HD. **d** Histograms of Cy5 PIFE ratio calculated as the mean Cy5 intensity in the light-gray region over that in the green region, yielding peaks at 1.98 ± 0.03, 1.75 ± 0.02, 1.75 ± 0.02, 1.83 ± 0.02 (SEM) for 0, 3.5, 5, and 10 nM Sen1 HD, respectively. **e** Duration histograms of a stalled Pol II remodeled by Sen1 HD (green zone) are globally fit to single-molecule Michaelis–Menten function with photobleaching correction (see "Methods" for details), yielding a binding rate constant of $k_+ = (6.8 ± 0.4) × 10^6 M^{-1} s^{-1}$ and $k_- ≈ 0$ with a reduced Chi-square of 1.0. $N = 337, 417,$ and 564 events were collected for Sen1 concentrations of 3.5, 5, and 10 nM, respectively. Source data are provided as a Source Data file.

## Sen1 dissociates with RNA simultaneously from Pol II TECs

To gain further insight into the mechanism of Sen1-dependent termination, we prepare a C-terminal SNAP tagged Sen1 HD and fluorescently label it with SNAP-surface 649 (Sen1 HD-SNAP649) and characterize the fate of Sen1 by monitoring SNAP649 fluorescence via single-molecule fluorescence assays. Two types of Pol II TECs containing an RNA-Cy3 are tethered to the PEG surface, via either biotinylated Pol II or biotinylated DNA (bioPol II/RNA-Cy3/DNA and Pol II/RNA-Cy3/bioDNA TECs, respectively, Supplementary Data 1), and subjected to ALEX modulation. The addition of 10 nM Sen1 HD-SNAP649, 2 nM TFIIS, 1.1 mM ATP, and 100 µM each of UTP and CTP, causes efficient transcription termination as reflected by a complete disappearance of Cy3 fluorescence at green laser excitation. Just before RNA-Cy3 disappearance, an RNA-Cy3/Sen1 HD-SNAP649 FRET signal is observed at green laser excitation, whose duration coincides with a discrete Sen1 HD-SNAP649 fluorescence pulse, reflecting a

single Sen1 HD-SNAP649 action, at red laser excitation (pink zone, 229 out of 247 events, 20 ms exposure time for each laser excitation, Fig. 2a and Supplementary Fig. 6). Although a majority of termination events display simultaneous dissociation of RNA-Cy3 and Sen1 HD-SNAP649, we observe 14 out of 247 Sen1 HD-SNAP649 fluorescence signals displaying a delayed disappearance relative to RNA-Cy3 disappearance with a mean duration of 1.8 ± 0.9 video frames (SEM, Fig. 2b; data collected at 50 Hz). Assuming that the simultaneous RNA-Cy3/Sen1 HD-SNAP649 dissociation events are detected due to the limited temporal resolution of our instrument, we can predict that the number of simultaneous dissociation events should be 11.8 ± 6.5, which is significantly lower than the number of 229 in our observation (Fig. 2b). Therefore, we can conclude that a majority of Sen1 HD-SNAP649 dissociates from the complex simultaneously with the RNA transcript. Those events for which Sen1 HD-SNAP649 disappearance is delayed compared to a loss of RNA-Cy3 could possibly

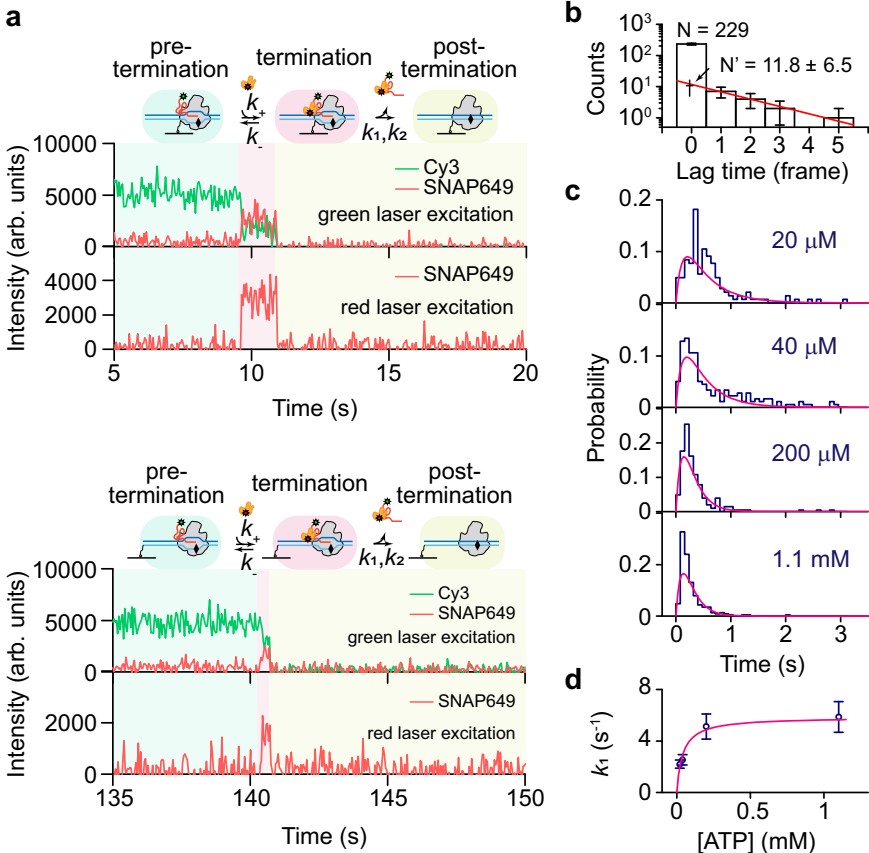

**Fig. 2 | Characterization of the formation and resolution of the Sen1-Pol II TEC intermediate. a** Typical trajectories showing Pol II termination mediated by Sen1 HD-SNAP649 (biotinylation on Pol II and on DNA in the upper and lower panels, respectively). Three zones are identified: a green zone, representing Pol II TEC initiating transcription and then stalling (i.e., the pre-termination state); a pink zone, representing the formation of a Sen1-Pol II TEC intermediate (i.e., the termination state) as reflected by the appearance of Sen1 HD-SNAP649 fluorescence in the red channel and its FRET-interactions with RNA-Cy3; and a yellow zone, representing a state after resolution of Sen1-Pol II TEC intermediate (i.e., the post-termination state). Simultaneous dissociation of Sen1 HD-SNAP649 and RNA-Cy3 are observed. **b** Histogram showing the number of events versus the lag time, in units of video frames (50 Hz acquisition frequency), between the disappearance of Sen1 HD-SNAP649 fluorescence and the disappearance of Cy3 fluorescence. Zero represents the simultaneous disappearance of Sen1 HD-SNAP649 and RNA-Cy3 ($N = 229$). Events which show a non-zero lag between disappearance events follow a single exponential decay giving an average of $1.8 \pm 0.9$ frames (SEM, $N = 14$) and a predicted number of $11.8 \pm 6.5$ (SEM) events at 0 frame. **c** Duration histograms of the Sen1-Pol II TEC intermediate at various ATP concentrations are fit to single-molecule Michaelis−Menten function (see "Methods" for details), giving $k_1 = 2.20 \pm 0.32$ (SEM, $N = 143$), $2.54 \pm 0.40$ (SEM, $N = 179$), $5.12 \pm 0.98$ (SEM, $N = 204$), and $5.87 \pm 1.19 \, \mathrm{s}^{-1}$ (SEM, $N = 229$) for 20 μM, 40 μM, 200 μM, and 1.1 mM ATP concentrations, respectively, and $k_2 = 8.69 \pm 2.03 \, \mathrm{s}^{-1}$ (SEM) with a reduced Chi-square of 1.8. A mean lifetime of $0.25 \pm 0.02$ s (SEM) is calculated for the 1.1 mM ATP condition. **d** The values for $k_1$ are further fit to a classical Michaelis−Menten model giving $k_1^{\max} = 5.9 \pm 1.0 \, \mathrm{s}^{-1}$ (SEM), and $K_m = 40 \pm 15$ μM (SEM, $N = 143$, 179, 204, and 229, respectively). Source data are provided as a Source Data file.

be due to its non-specific interaction with biotinylated Pol II, biotinylated DNA or the PEG surface. As the above measurements are performed on Pol II TECs tethered to a PEG surface via either biotinylated Pol II or biotinylated DNA (Fig. 2a and Supplementary Fig. 6), these results reflect a complete separation of Sen1 HD-SNAP649/RNA-Cy3 from tethered Pol II/DNA on the surface. Overall, we conclude that Sen1 HD-SNAP649 releases and dissociates with the RNA transcript simultaneously and completely from Pol II TECs during Sen1-dependent termination.

## Sen1 translocates on RNA to target Pol II TEC and hydrolyzes a single ATP to dissociate RNA transcript

The presence of a Cy3/SNAP649 FRET signal before termination reflects the formation of Sen1-Pol II TEC intermediate containing a single Sen1 HD-SNAP649 molecule[15,17], which possibly corresponds to the catalytic step of the Sen1-dependent termination mentioned in the first section. The mean duration of the intermediate is $0.25 \pm 0.02$ s at 1.1 mM ATP (Fig. 2c) and its inverse can be used as the $k_{cat}$ to interpret the binding and dissociation kinetics of Sen1 HD in solution by fitting the distributions of Sen1 HD termination time to single-molecule Michaelis−Menten

function (Eq. (4)) giving binding rate of $k_+ = (6.8 \pm 13.2) \times 10^6 \, \mathrm{M}^{-1} \mathrm{s}^{-1}$ and dissociation rate of $k_- = (-1.9 \pm 7.8) \times 10^{-16} \, \mathrm{s}^{-1}$. When constraining $k_- = 0$, we get $k_+ = (6.8 \pm 0.4) \times 10^6 \, \mathrm{M}^{-1} \mathrm{s}^{-1}$, the same as that from the earlier fit to a single exponential function (Fig. 1e, Supplementary Fig. 4c, Supplementary Notes, and Supplementary Data 2).

Duration histograms of Cy3/SNAP649 FRET displaying a fast rise and a slow exponential decay reflect the existence of two sequential steps (Fig. 2c)[36]. Titrating ATP concentration from 20 μM to 1.1 mM significantly enhances the rate for the slow exponential decay, reflecting the existence of an ATP-dependent step. Since the rate of the slow decay is not linearly proportional to ATP concentration, an ATP-independent step must exist[37]. By globally fitting these duration histograms to a two-step model simplified from the single-molecule Michaelis−Menten function (Eq. (5))[36], we can characterize these two steps and obtain an ATP-dependent rate parameter $k_1$ describing the ATP-dependent translocation activity of Sen1 which contains multiple ATP binding/hydrolysis steps and is consistent with the prior report that RNA transcript and ATP hydrolysis are essential for Sen1-dependent termination[15]. An ATP-independent behavior of $k_2$ indicates a single ATP hydrolysis is essential and sufficient for the

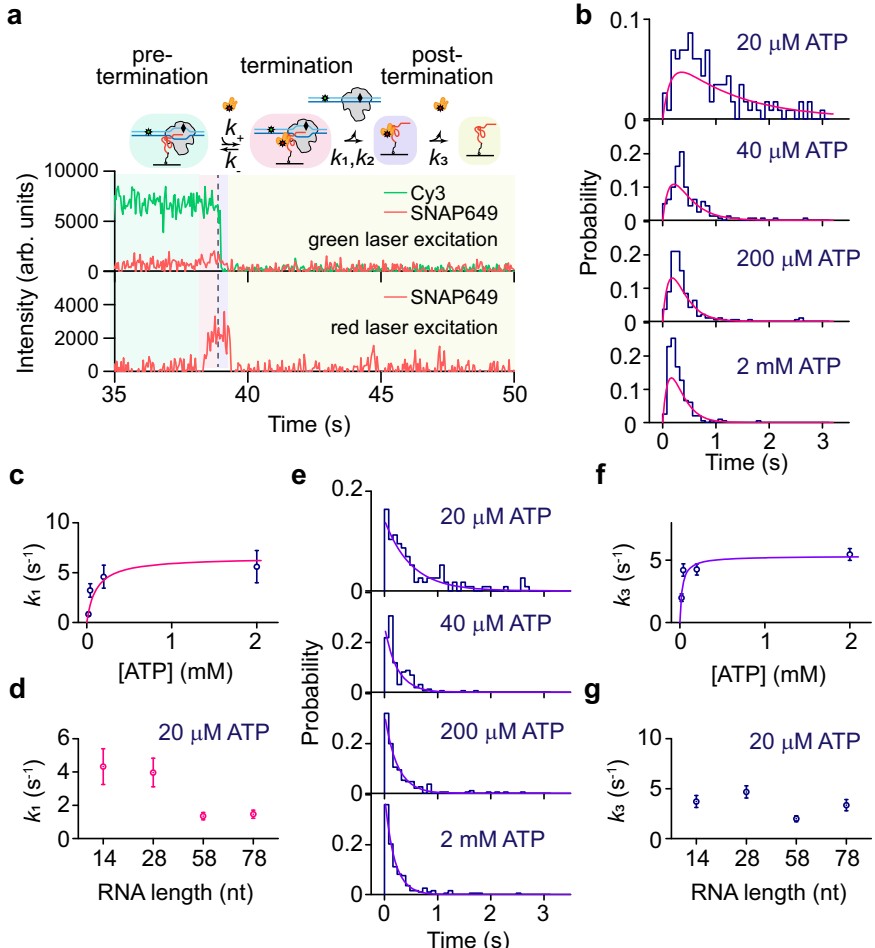

**Fig. 3 | Characterization of Sen1-SNAP649 action on Pol II TEC tethered via a biotinylated RNA. a** Typical trajectories showing transcription termination of Pol II/bioRNA/DNA$_{58bp}$-Cy3 TECs by Sen1 HD-SNAP649. A post-termination state showing persistent interaction of Sen1 HD-SNAP649 with RNA as reflected by a lag time between disappearance of the DNA$_{58bp}$-Cy3 signal (purple zone) compared to those in Fig. 2a. **b** Lifetime distributions of the Sen1-Pol II TEC intermediate (pink zone) are globally fit to single-molecule Michaelis–Menten function (see "Methods") giving $k_1 = 0.84 \pm 0.19$, $3.21 \pm 0.68$, $4.59 \pm 1.15$, and $5.60 \pm 1.61 \, s^{-1}$ (SEM, $N = 116$, 160, 190, and 197 for 20 μM, 40 μM, 200 μM, and 2 mM ATP concentrations, respectively) and $k_2 = 6.71 \pm 2.01 \, s^{-1}$ (SEM) with a reduced Chi-square of 1.8. **c** $k_1$ values are fit to classical Michaelis–Menten model giving $k_1^{max} = 6.5 \pm 1.6 \, s^{-1}$ (SEM), and $K_m = 109 \pm 41 \, \mu M$ (SEM, $N = 116$, 160, 190, and 197, respectively). **d** $k_1$ values (mean ± SEM, $N = 170$, 195, 116, and 118, respectively) obtained using various RNA lengths at 20 μM ATP (see Supplementary Fig. 8a) are plotted versus the length of RNA that extends from the RNA exit channel of Pol II. **e** Lifetime distributions of Sen1 HD-SNAP649 residence in the post-termination state (purple zone) as a function of ATP concentration are individually fit to a single exponential function, giving a rate constant ($k_3$) of $1.99 \pm 0.30$, $4.21 \pm 0.50$, $4.25 \pm 0.45$, and $5.46 \pm 0.48 \, s^{-1}$ (SEM). **f** The rate constants from the prior panel follow a classical Michaelis–Menten model with $k_3^{max} = 5.33 \pm 0.40 \, s^{-1}$ (SEM), and $K_m = 27 \pm 7 \, \mu M$ (SEM, $N = 116$, 160, 190, and 197, respectively). **g** Dissociation rate constant (mean ± SEM, $N = 170$, 195, 116, and 118) of Sen1 HD-SNAP649 in the post-termination state versus the length of RNA that extends from the RNA exit channel of Pol II (see Supplementary Fig. 8b). Source data are provided as a Source Data file.

catalytic termination of Sen1/RNA release (see "Methods" and Fig. 2c). The possibility of multiple ATP binding events can be excluded because Sen1 has only one ATP binding site[38]. $k_1$ values follow the classical Michaelis–Menten model, giving a maximum rate of $k_1^{max} = 5.9 \pm 1.0 \, s^{-1}$, and $K_m = 40 \pm 15 \, \mu M$ (Fig. 2d). Taking into account the transcription unit of 28 bp and a 16-mer RNA primer of Pol II TECs, an RNA with a total length of ~44 nt will be generated, resulting in ~28-nt RNA extending from the RNA exit channel of Pol II[39]. A mean translocation rate of ~83 nt s$^{-1}$ for Sen1 HD-SNAP649 on RNA at saturating ATPs can be estimated by multiplying $k_1^{max}$ by 14 nt, assuming that Sen1 HD-SNAP649 most probably (i.e., on average) initiates translocation from the center of the ~28-nt extended RNA.

The kinetic features of $k_1$ and $k_2$ are verified and confirmed with another version of Pol II TEC tethered through biotinylated RNA (Pol II/bioRNA$_{16}$/DNA$_{58bp}$-Cy3 TECs, Supplementary Data 1). Upon addition of 10 nM Sen1 HD-SNAP649, 2 nM TFIIS, 20 μM ATP, and 1 mM each of UTP and CTP, we again observe a complete extinction of the

DNA$_{58bp}$-Cy3 signal coinciding with a Sen1 HD-SNAP649 fluorescence pulse, but a delayed disappearance of Sen1 HD-SNAP649 fluorescence under ALEX illumination (Fig. 3a and Supplementary Fig. 7). The time during which DNA$_{58bp}$-Cy3 and Sen1 HD-SNAP649 signals overlap (pink zone) reflects the time required for Sen1 to complete its task, and its statistical analysis reveals consistent results (Fig. 3b, c). The time required for Sen1 HD to terminate Pol II transcription (pink zone in Fig. 3a) is fit to a two-step model (Eq. (5)) and the extracted $k_1$ values display a dependence on the length of the RNA transcript (Fig. 3d and Supplementary Fig. 8a). To exclude the possibility that different $k_1$ values are due to the different DNA lengths, we characterize the ATPase activity of Sen1 HD on linearized DNA, single-stranded DNA, and single-stranded RNA. We find rare ATPase activity on linearized DNA but significantly higher ATPase activity on either single-stranded DNA or RNA (Supplementary Fig. 9). We conclude that $k_1$ reflects a translocation action of Sen1 HD-SNAP649 along RNA transcript to form the Sen1-Pol II TEC intermediate.

## Sen1 hydrolyzes ATPs on RNA after their simultaneous dissociation

Because the NNS-dependent termination pathway is believed to be directly coupled with the RNA degradation complex[40], we next characterize Sen1 action on RNA after their simultaneous dissociation from Pol II TECs. Since Sen1 is a helicase that can unwind RNA and DNA duplex or RNA/DNA hybrids[16,41], we expect that Sen1 may remain and translocate on RNA after their simultaneous dissociation from the TEC. To test this idea, we prepare Pol II/bioRNA$_{16}$/DNA$_{58bp}$-Cy3 TECs (Supplementary Data 1) bearing a 58 bp transcription unit and tether them to the PEG surface via biotinylated RNA. The termination will release Pol II and DNA as characterized earlier while the RNA transcript remains localized to the PEG surface, allowing continuous detection of Sen1 action after TEC dissociation. Upon addition of 10 nM Sen1 HD-SNAP649, 2 nM TFIIS, 20 μM ATP, and 1 mM each of UTP and CTP, we observe an overlap of DNA-Cy3 fluorescence and Sen1 HD-SNAP649 fluorescence, followed by DNA-Cy3 disappearance, itself followed by Sen1 HD-SNAP649 disappearance (purple zone, Fig. 3a and Supplementary Fig. 7). This delayed Sen1 HD-SNAP649 disappearance indicates Sen1 HD remains on RNA after termination. The distribution of delay times follows a single exponential decay suggesting a Poisson process for Sen1 HD-SNAP649 dissociation, and its mean value or rate constant ($k_3$) varies with ATP concentration, which suggests ATP hydrolysis activity and might relate to a translocation activity of Sen1 HD-SNAP649 on the released RNA in the post-termination state (Fig. 3e). These $k_3$ constants can be described by the classical Michaelis–Menten model yielding $k_3^{max} = 5.33 \pm 0.40\,s^{-1}$, and $K_m = 27 \pm 7\,\mu M$ (Fig. 3f). Assuming that Sen1 HD-SNAP649 contacts Pol II to induce termination[16] and translocates afterwards, the RNA length downstream of Sen1 HD-SNAP649 after termination could be considered as the length of RNA occupied by Pol II, i.e., ~16 nt[39]. An independence of the duration of Sen1 HD-SNAP649 action on the RNA length that extends from the RNA exit channel of Pol II (Fig. 3g and Supplementary Fig. 8b) suggests no backward translocation of Sen1 after termination. Thus, we propose that after termination Sen1 continues translocating along the ~16 nt RNA in front of it with a translocation rate at saturating ATP estimated as ~16 nt*5.33 s$^{-1}$ = ~85 nt s$^{-1}$. This is consistent with the earlier characterization of the translocation rate (~83 nt s$^{-1}$) of Sen1 HD-SNAP649 leading up to the formation of the Sen1-Pol II TEC intermediate.

## Pol II dissociates from, retains to, or diffuses along DNA after Sen1/RNA release

As has been previously characterized that *E. coli* RNAP can diffuse along DNA after intrinsic termination[35,42], it would be interesting to examine the fate of Pol II after Sen1-dependent termination. We prepare Pol II TECs with RNA-Cy3, biotinylated Pol II, and DNA-Cy5(+3) (bioPol II/RNA-Cy3/DNA-Cy5(+3), Supplementary Data 1), for which the dissociation of RNA-Cy3 reflects transcription termination and that of DNA-Cy5 reflects Pol II kinetics after termination. Upon addition of 10 nM Sen1 HD, 2 nM TFIIS, 2 mM ATP, and 1 mM each of UTP and CTP with ALEX illumination, efficient RNA release is observed simultaneously with Pol II dissociation from DNA in about 1/3 of events (58 out of 170 events, typical trajectories shown Fig. 1b) consistent with prior reports[15]. Moreover, we also observe Pol II retained to the DNA after termination (Fig. 4a), with 74/170 Pol II molecules dissociating from DNA after a mean time ($1/k_4$) of $36 \pm 9\,s$, and 34/170 Pol II molecules retained throughout the duration of the recording (Fig. 4b, upper panel). The extra addition of 10 μg ml$^{-1}$ heparin slightly increases the fraction of simultaneous Pol II/RNA dissociations but significantly reduces the mean dissociation time of retained Pol II ($2.1 \pm 0.3\,s$, SEM, Fig. 4b, c), which is also true when replacing heparin with an excess of unlabeled DNA (Supplementary Fig. 10). These results suggest a primary interaction of Pol II with the DNA backbone in the post-termination state rather than with DNA bases as in the elongation state,

which implies in return a fully rewound transcription bubble after termination. By positioning Cy5 from the +3 position to the 3' or the 5' end of the non-template DNA in Pol II TECs to generate different distances between Cy5 and termination site (bioPol II/RNA-Cy3/DNA10-Cy5(3'), bioPol II/RNA-Cy3/DNA19-Cy5(3'), bioPol II/RNA-Cy3/DNA49-Cy5(3'), bioPol II/RNA-Cy3/DNA61-Cy5(5'), and bioPol II/RNA-Cy3/DNA91-Cy5(5'), respectively, Supplementary Data 1), Cy5 PIFE is observed immediately after RNA release but not before termination (Fig. 4f and Supplementary Fig. 11). This possibly suggest Pol II capturing the DNA end where Cy5 locates because Pol II is believed to have a high affinity for DNA ends[35]. Analysis on the lifetime and the fraction of Cy5 PIFE appearance after termination better supports a one-dimensional (1D) diffusion model yielding the diffusion coefficient of $(1.3 \pm 0.3) \times 10^{-4}\,\mu m^2\,s^{-1}$ for Pol II post-termination and the fraction constant of $0.26 \pm 0.02$ (dark blue curve for 1D diffusion model and light blue for 3D diffusion model in Fig. 4g and Supplementary Notes). However, when Cy5 is placed 539 bp away from the termination site, no Cy5 PIFE can be observed from 116 termination events suggesting a short range of Pol II diffusion compared to that of *E. coli* RNAP[42] (Supplementary Fig. 11e). To exclude the possibility that the observation of no Cy5 PIFE is not due to inactive Pol II TECs on this long construct, the elongation activity of the freshly prepared Pol II TECs is measured by adding NTPs. Since Pol II can only reach DNA end via transcription through the 539 bp DNA length, the elongation time can be estimated as the time from NTPs addition until the appearance of Cy5 PIFE reflecting Pol II reaching DNA end, yielding a mean lifetime of $47 \pm 24\,s$ (Supplementary Fig. 11f, g). Pol II transcription rate can be estimated as 539 bp/47 s = 11.5 bp s$^{-1}$ supporting Pol II elongation activity and is consistent with previous reports[17,22]. The efficiency of Pol II restarting transcription (51 out of 104 Pol II molecules) is observed consistent with the fraction obtained in Fig. 1c, again supporting active Pol II TECs on the long DNA construct. Levels of Cy5 PIFE when labeling at the downstream (-1.4) and upstream (-1.7) ends of the DNA are presented (Fig. 4h).

## Rewinding of the transcription bubble facilitates the resolution of Sen1-Pol II TEC intermediate

To examine the effect of the rewinding of the transcription bubble on Pol II dissociation during termination, a 10-base mismatch is introduced at the downstream end of G-less cassette to form a 10-base mismatch bubble for a stalled Pol II TEC (bioPol II/RNA-Cy3/DNA$_{+19\sim+28mis}$-Cy5(+3), Supplementary Data 1). By subjecting these Pol II TECs to single-molecule fluorescence assays, we observe a slightly reduced termination efficiency as characterized by the fraction of RNA release from the total number of elongated and stalled Pol II TECs (Fig. 4i), but rare Pol II dissociation events coincided with RNA release (5 out of 166 events, Fig. 4d, e). This fraction is barely affected by heparin (10 μg ml$^{-1}$ final concentration). Retained Pol II molecules dissociate more slowly from DNA ($89 \pm 11\,s$ in the absence of heparin and $72 \pm 10\,s$ in the presence of 10 μg ml$^{-1}$ heparin) than if the transcription bubble is fully complementary. These results suggest that the retained Pol II molecule possibly remains at the mismatched transcription bubble which cannot be competed off by heparin. Therefore, we conclude that rewinding of the transcription bubble is necessary for Pol II dissociation but is insufficient.

To more deeply probe the effects of transcription bubble rewinding on the resolution of the Sen1-Pol II TEC intermediate, we engineered the non-template strand of the downstream end of the G-less cassette while maintaining the template sequence unchanged to generate a partially or completely mismatched bubble for a stalled Pol II (Fig. 5a). These mismatches will increase the energy barrier for the rewinding of the transcription bubble during Pol II termination. Sen1-dependent Pol II termination is again observed on all of these constructs by subjecting the corresponding TECs (Pol II/RNA-Cy3/bioDNA$_{+19\sim+28mis, +19\sim+23mis, +24\sim+28mis}$, Supplementary Data 1) and Sen1

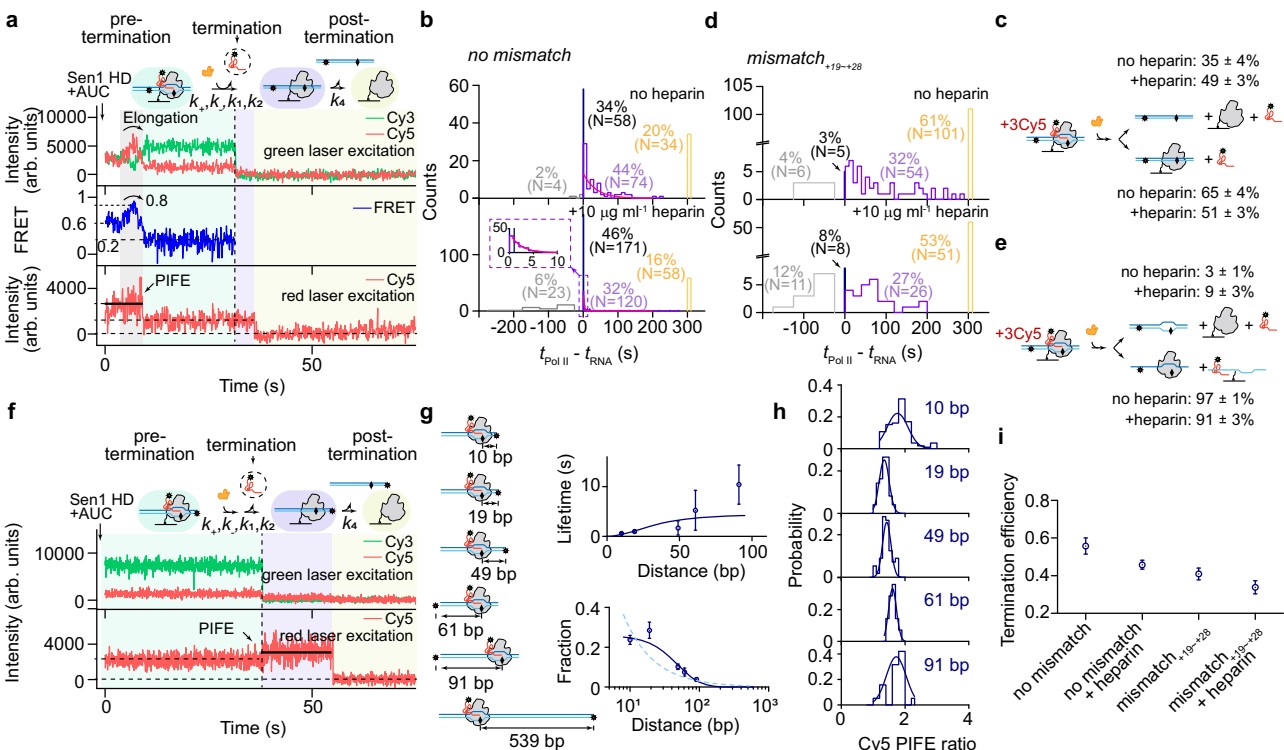

**Fig. 4 | Characterization of Pol II dissociation from DNA in the post-termination state. a** Typical termination trajectories showing sequential dissociation of RNA (Cy3 disappearance, dash line) and Pol II (Cy5 disappearance) on bioPol II/RNA-Cy3/DNA-Cy5(+3) TECs. **b** Histograms of the time elapsed between RNA release and Pol II dissociation under the effect of 10 μg ml⁻¹ heparin, i.e., Pol II dissociates before (gray), simultaneously with (navy), or after (purple) RNA release. Orange represents Pol II retaining on DNA over the course of the recording. The mean times ($1/k_4$) of purple events are $36 \pm 9$ s (SEM, $N = 74$) and $2.1 \pm 0.3$ s (SEM, $N = 120$), respectively. **d** As for (**b**) but obtained on bioPol II/RNA-Cy3/DNA$_{+19\sim+28\text{mis}}$-Cy5(+3) TECs with mismatches at $+19\sim +28$ yielding increased dissociation times of $89 \pm 11$ s (SEM, $N = 54$) and $72 \pm 10$ s (SEM, $N = 26$), respectively. Schematic of Pol II dissociation pathways during Sen1-dependent termination on bioPol II/RNA-Cy3/DNA-Cy5(+3) TECs (**c**) and under the effect of DNA mismatches

(**e**). **f** Typical trajectories when labeling Cy5 at the downstream DNA end (bioPol II/RNA-Cy3/DNA-Cy5(3′) TECs, 49/172 events). **g** The mean lifetimes of Cy5 PIFE appearance (upper panel) and its fraction (lower panel) versus different distances of Cy5 fluorophore from the termination site (mean ± SEM, $N = 112, 49, 56, 14$, and 19 respectively). Data better fit to one-dimensional diffusion[35] model yielding diffusion coefficient of $(1.3 \pm 0.3) \times 10^{-4}$ μm² s⁻¹ (SEM) and the fraction of $0.26 \pm 0.02$ (SEM, a reduced chi-square of 1.2 for 1D model, dark blue line, compared to that of 7.1 for 3D model, light-blue dash line). **h** Corresponded histograms of Cy5 PIFE ratio give Gaussian peaks at $1.77 \pm 0.04$, $1.35 \pm 0.02$, $1.43 \pm 0.02$, $1.61 \pm 0.04$, $1.72 \pm 0.08$ (SEM), respectively. **i** Termination efficiencies (mean ± SEM) of engineered Pol II TECs with no mismatch (170/350), mismatch (166/406) in the absence of heparin, and no mismatch (372/812), mismatch (96/284) in the presence of 10 μg ml⁻¹ heparin. Source data are provided as a Source Data file.

HD-SNAP649 to single-molecule fluorescence assays. Nearly all termination events (106 out of 106, 115 out of 115, 149 out of 150 and 156 out of 164 events for Pol II TECs with no mismatch, mismatch$_{+19\sim+28}$, mismatch$_{+19\sim+23}$ and mismatch$_{+24\sim+28}$, respectively) show simultaneous dissociation of Sen1 HD-SNAP649 and RNA transcript, suggesting the simultaneous dissociation of Sen1/RNA is independent of mismatches in the transcription bubble. However, when the lifetime of Sen1-Pol II TEC intermediates is analyzed, a gradual increase in the mean duration for Pol II termination is observed when increasing the size of the mismatch sequence in the transcription bubble (Fig. 5b). Although we have shown earlier that the duration of Sen1 HD-SNAP649 fluorescence or RNA-Cy3/Sen1 HD-SNAP649 FRET involves both formation and resolution of the Sen1-Pol II TEC intermediate, it is reasonable to propose that the rewinding of the transcription bubble affects only the resolution step, where a conformational change of the transcription bubble may be required[43], rather than the formation of Sen1-Pol II TEC intermediate. Fitting the duration histograms to the two-step model (Eq. (5)) yields $k_1 = 4.27 \pm 1.11$ s⁻¹ and statistically indistinguishable $k_2$ values (Fig. 5c, d). However, these $k_2$ values show a weak correlation with that of the mean duration in Fig. 5b. These results suggest that rewinding of the transcription bubble facilitates resolution of the Sen1-Pol II TEC intermediate, i.e., RNA release during Sen1-dependent termination, yet incomplete dissociation of Pol II from DNA.

## Discussion

We report a comprehensive set of assays characterizing Sen1-dependent transcription termination at single-molecule resolution (Fig. 6). To remodel a stalled Pol II, Sen1 binding to and translocation along the nascent RNA transcript is indeed required. Sen1 binding to the RNA transcript is unlikely to be completely random. Indeed in the NNS-dependent pathway Nrd1 and Nab3 specifically bind to conserved motifs of RNA transcripts and are believed to be functional for the recruitment of Sen1 to Pol II TECs[11,12]. Sen1 may be recruited to the RNA transcript by these two factors to accelerate the process of Sen1-dependent termination. We estimate a mean rate of ~80 nt s⁻¹ for Sen1 translocating on RNA, which is much higher than that of Pol II transcription as reported[17,22,30]. This elevated translocation rate enables a kinetic competition between Sen1-dependent termination and Pol II transcription as proposed[19]. To terminate Pol II transcription, a termination intermediate between Sen1 and Pol II TEC is potentially formed[15,17], and here, we reveal the presence of a single Sen1 molecule in the intermediate as visualized by single-molecule fluorescence assays. Formation of the termination intermediate requires Sen1 translocation along the RNA transcript by hydrolyzing multiple ATPs[15], while its resolution step requires the hydrolysis of a single ATP molecule, which is sufficient to simultaneously dissociate Sen1 and the RNA transcript. Sen1 subsequently remains translocating along the RNA transcript until complete separation from the RNA. As the NNS-

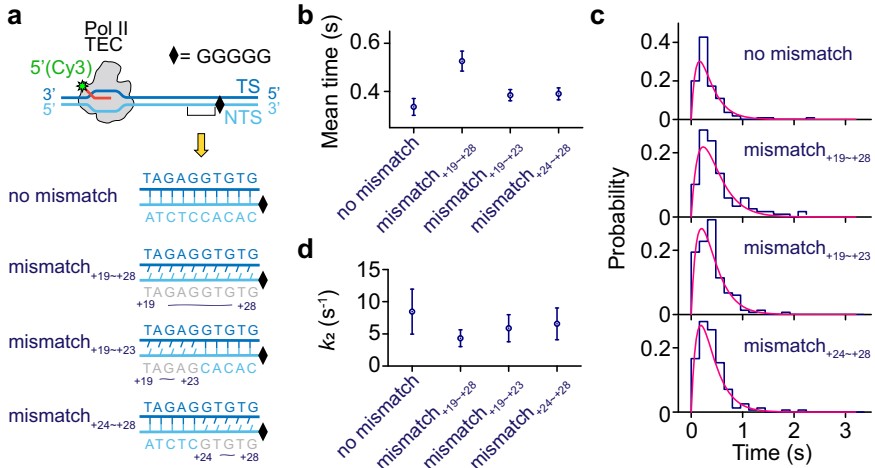

**Fig. 5 | Role of transcription bubble rewinding on Sen1-dependent termination.**
**a** Schematic of mismatched regions (+19 to +28, +19 to +23, and +24 to +28) implemented immediately upstream of the G-stretch. **b** Mean durations of Sen1-Pol II TEC intermediate prior to termination are analyzed: $0.34 \pm 0.03$, $0.53 \pm 0.04$, $0.38 \pm 0.02$, and $0.39 \pm 0.02$ s (SEM, $N = 110$, 118, 149, and 156 for Pol II TECs with no mismatch, mismatch$_{+19\sim+28}$, mismatch$_{+19\sim+23}$ and mismatch$_{+24\sim+28}$, respectively). **c** Lifetimes of Sen1-Pol II TEC intermediate prior to termination are plotted and

globally fit to the single-molecule Michaelis–Menten model, yielding $k_1 = 4.27 \pm 1.11$ s$^{-1}$ and $k_2 = 8.45 \pm 3.49$, $4.33 \pm 1.31$, $5.88 \pm 2.09$, and $6.56 \pm 2.47$ s$^{-1}$ (SEM, $N = 110$, 118, 149, and 156 for Pol II TECs with no mismatch, mismatch$_{+19\sim+28}$, mismatch$_{+19\sim+23}$ and mismatch$_{+24\sim+28}$, respectively) with a reduced Chi-square of 1.3. **d** $k_2$ (mean $\pm$ SEM) are plotted for the four cases ($N = 110$, 118, 149, and 156). Source data are provided as a Source Data file.

dependent termination pathway is believed to be directly coupled with the exosomal RNA degradation complex[40], Sen1 retention on the released RNA transcript may have some relation to the RNA degradation pathway. In our results, Sen1 acts like a translocase moving along the RNA transcript and removing Pol II as an obstacle in its path. Although the Sen1-dependent termination activity requires specific interactions between Sen1 and Pol II, our results provide strong evidence for the previously proposed rho-like model for Sen1-dependent termination[15]. This extrinsic termination model is analogous to that observed with Sen1's functional homolog in *E. coli*−ρ helicase, for which a termination intermediate has also been characterized[21].

As Sen1 releases the RNA transcript, the transcription bubble rewinds to increase the termination efficiency which has been previously characterized by the amount of RNA released from Pol II TEC[16]. In this study, we find that transcription bubble rewinding slightly enhances the kinetics of Sen1/RNA dissociation, and is absolutely required for Pol II dissociation from the DNA by direct observing the separation of Pol II from DNA after RNA release. Consistent with this, a negatively supercoiled DNA, which disfavors the rewinding of the transcription bubble, slows down the resolution of the termination intermediate[17]. However, transcription bubble rewinding cannot completely result in Pol II dissociation from the DNA. We observe that around one third of Pol II molecules dissociate simultaneously with RNAs (Fig. 4b) consistent with prior report[15], but more intriguingly, of the two-thirds of retained Pol II molecules, about half undergo 1-D diffusion along the DNA. *S. cerevisiae* Pol II possesses a comparable diffusion coefficient but cannot diffuse as far as *E. coli* RNA polymerase[35,42]. This difference might be related to the chromatinized DNA in *S. cerevisiae* but not in *E. coli*, and the biological relevance remains elusive.

## Methods
### Plasmids for protein purification
Sen1 HD (residues 1095–1904) was amplified from Sen1-CPD-His8 plasmid[38] using a pair of PCR primers: C_Sen1HD_F_NcoI and C_Sen1HD_R_SpeI (Supplementary Data 3). The PCR product was digested with NcoI and SpeI (New England Biolabs) restriction enzymes and purified with a PCR and Gel extraction kit (Macherey-

Nagel). The purified product was then subcloned into a modified pGEX6P-1(ΔGST) vector[44] where a SpeI restriction site was previously incorporated at the N-terminus of the PreScission cleavage site using site-direct mutagenesis (QuickChange kit, Agilent). Sen1 HD thus carries in the C-terminus a PreScission cleavage site followed by a SNAP-tag and a 10×His-tag which was used for the purifications of Sen1 HD and Sen1 HD-SNAP.

### Protein purification
*S. cerevisiae* RNA Pol II was purified from BJ5464 strain that expresses a His6-tagged version of Rpb3 essentially as previously described[38]. Briefly, the cell pellet was resuspended in lysis buffer (20 mM Tris-HCl pH 8, 150 mM KCl, 10% (v/v) glycerol, 10 μM ZnCl$_2$, 10 mM DTT) and lysed using a MM400 (Retsch). After clarification, the protein extract was subjected to Ni-affinity chromatography (Ni-NTA, Qiagen) and then anion exchange chromatography (Mono-Q 5/50 GL, GE Healthcare). The fractions of interest were concentrated and dialyzed against Pol II storage buffer (10 mM HEPES pH 7.9, 40 mM (NH4)$_2$SO$_4$, 10 μM ZnCl$_2$, 10% (v/v) glycerol, 10 mM DTT) and stored at −80 °C.

Biotinylated version of Pol II was prepared by incubating 50 μg of Pol II bearing the AviTag and 10 μg of BirA biotin ligase protein in 200 μl reaction buffer (10 mM HEPES pH 7.9, 40 mM (NH4)$_2$ SO$_4$, 5 μM ZnCl$_2$, 2.5 mM DTT, 5% (v/v) glycerol, 10 mM ATP and 0.1 mM biotin) for 5 hours at 4 °C. Free components were removed by dialyzing against Pol II storage buffer and stored at −80 °C.

Sen1 HD was prepared as described in ref. 38 with some modifications. Briefly, Sen1 HD with a cleavable C-terminal SNAP-tag and His-tag was expressed in *E. coli* BL21(DE3) codon plus RIPL (stratagene) cells. Cell pellet was resuspended in 20 mM sodium phosphate pH 8.0, 500 mM NaCl, 2 mM MgCl$_2$, 30 mM imidazole, 10% (v/v) glycerol, 1 mM β-mercaptoethanol, benzonase, and protease inhibitors. The lysate was clarified and subjected to Ni$^{2+}$ affinity chromatography (HisTrap HP, GE Healthcare). After overnight cleavage with PPX, the elute was subtracted with HisTrap HP column and then further purified with anion exchange chromatography (HiTrap Heparin HP, GE Healthcare) in 20 mM Tris-HCl pH 7.5, 200 mM NaCl, 2 mM MgCl$_2$, 1 mM DTT. The elute was obtained by increasing NaCl gradient to 1 M and then concentrated for size-exclusion chromatography (HiLoad superdex200,

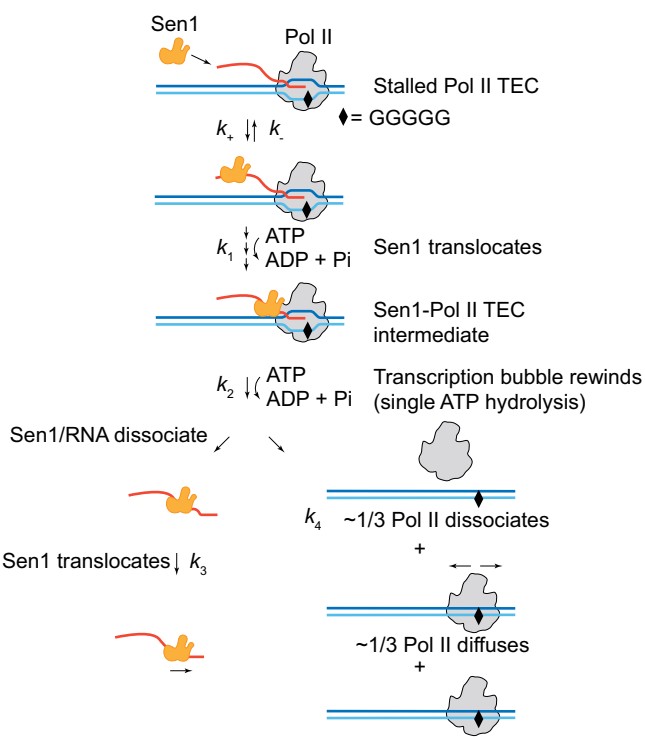

**Fig. 6 | Model for Sen1-dependent termination of Pol II transcription.** Sen1 diffuses in solution to bind to Pol II TEC ($k_+$) and then translocates along the RNA transcript at ~80 nt s$^{-1}$ until it reaches the Pol II, forming a Sen1-Pol II TEC intermediate. This intermediate hydrolyzes a single ATP ($k_2$) to dissociate Sen1 and RNA from the complex, resulting in release of only approximately 1/3 of Pol II molecules from the DNA. The time required for Sen1 to cause termination increases if the transcription bubble cannot be rewound. Sen1 then translocates along the released RNA transcript (at ~80 nt s$^{-1}$) until complete dissociation. Of the 2/3 Pol II TECs retained to DNA after termination, about half undergo 1D diffusion along DNA in the post-termination state.

GE Healthcare) in 20 mM HEPES pH 7.5, 300 mM NaCl, 2 mM MgCl$_2$, and 1 mM DTT, 10% (v/v) glycerol. The purified proteins were aliquoted and stored at −80 °C.

Sen1 HD-SNAP was prepared following the same procedure as Sen1 HD, except that the anion exchange chromatography was performed directly after Ni2+ affinity chromatography without the PPX cleavage step.

TFIIS was purified as described[45]. Briefly, cell lysate was purified with Ni$^{2+}$-affinity chromatography (HisTrap HP, GE Healthcare) and then with anion exchange chromatography (Mono-S 5/50 GL, GE Healthcare). The second peak fraction from Mono-S column was concentrated and then further purified with size-exclusion chromatography (HiLoad superdex200, GE Healthcare). Purified proteins were aliquoted and stored at −80 °C.

His-tagged BirA was expressed in *E. coli* BL21(DE3) cells (Novagen) grown in 2-l culture. Briefly, cell pellet was resuspended in 25 mM Tris-HCl pH 8, 250 mM NaCl, 5% (v/v) glycerol, 1 mM β-mercaptoethanol. After clarification, the lysate was subjected to Ni$^{2+}$-affinity chromatography (HisTrap HP, GE Healthcare) and eluted with an imidazole gradient to 1 M. Purified proteins were aliquoted and stored at −80 °C.

GST-PPX (PreScission protease) was expressed in *E. coli* BL21(DE3) cells (Novagen) grown in 1-liter culture. Briefly, cell pellet was resuspended in 25 mM Tris-HCl pH 7.5, 0.5 M NaCl, 5 mM DTT and the lysate was loaded onto a 5 ml GST column (GE Healthcare). After washing with 400 ml buffer thoroughly, protein of interest was eluted in 50 mM Tris-HCl pH 8.2, 0.2 M NaCl, 5 mM DTT, 20 mM reduced glutathion (Merck) and stored at −80 °C.

## Sen1 HD-SNAP labeling

SNAP-Surface 649 (New England Biolabs) was dissolved in DMSO at 1 mM concentration for stock. Sen1 HD-SNAP proteins were mixed with a fivefold excess of SNAP-Surface 649 and 2 mM DTT in Sen1 HD-SNAP storage buffer (20 mM HEPES pH 7.5, 300 mM NaCl, 2 mM MgCl$_2$, and 1 mM DTT, 10% (v/v) glycerol). The mixture was incubated at 4 °C overnight and then separated via size-exclusion chromatography (HiLoad superdex200, GE Healthcare). Proteins were aliquoted and stored at −80 °C. All these procedures were performed in the dark.

## Preparation of Pol II TECs

For Pol II TEC preparation, RNA and DNA templates were mixed at equal molar ratios in T50 buffer (10 mM Tris-HCl pH 8.0, 50 mM NaCl in DEPC-treated water) at a final concentration of 5 μM. The mixture was heated to 95 °C for 5 min, followed by fast cooling to 65 °C and then slowly cooling at a rate of 1 °C per min to 4 °C[39]. The mixture was then mixed with 0.5 molar excess of Pol II or biotinylated Pol II, 2.5 μl RNase inhibitor (M0314S, New England Biolabs) in transcription buffer (20 mM HEPES pH 8.0, 100 mM KCl, 8 mM MgCl$_2$, 0.5 mg ml$^{-1}$ BSA, 7.5 μM ZnCl$_2$, 2 mM β-mercaptoethanol) and incubated at 25 °C for 20 min with shaking at 550 rpm. Two molar excess of non-template DNA relative to the RNA/DNA hybrid was added and incubated at 25 °C for 20 min with shaking at 550 rpm. To remove the unconstructed nucleic acids, the mixture was incubated with 20 μl prewashed Ni-NTA magnetic beads (New England Biolabs) in the presence of 15 mM imidazole at 4 °C for 20 min with shaking at 550 rpm. Free components were removed by discarding the supernatant. Added 200 μl TB buffer containing 15 mM imidazole and 2.5 μg ml$^{-1}$ heparin (Sigma-Aldrich) and incubated at 4 °C for 5 min with shaking at 550 rpm. Discarded the supernatant and repeated the wash step four times. Pol II TECs were eluted from beads with 200 μl TB buffer containing 200 mM imidazole by incubating at 4 °C for 20 min with shaking at 550 rpm. The elute can be tethered on streptavidin-coated PEG surface for single-molecule fluorescence experiments. Reconstructed Pol II TECs for different experimental purposes are summarized in Supplementary Data 1.

It is worth noting that the bioPol II/RNA-Cy3/DNA539-Cy5(3′) complex is firstly assembled with short DNA oligos and then ligated to a PCR-amplified dsDNA with 5′ Cy5 labeling. This gives a distance of 539 bp from Pol II stalling site to the Cy5 fluorophore. The 5′ Cy5-labeled dsDNA is prepared with a pair of primers: 539 F and 539R-Cy5(5′), followed by purification from agarose gel migration (Macherey-Nagel PCR and Gel Extraction Kit). The dsDNA is digested 3 h with SbfI restriction enzyme (New England Biolabs), gel purified and then ligated to the bioPol II/RNA-Cy3/DNA539-Cy5(3′) complex with a molar ration of 3:1 (1 represents the amount of Pol II molecules) at 16 °C for 30 min. The ligation product is tethered to the glass surface precoated with PEG and streptavidin. The complexes are washed with transcription buffer containing 2.5 μg ml$^{-1}$ heparin to ensure the removal of T4 ligase from DNA.

## Single-molecule fluorescence assay

Single-molecule fluorescence assay was performed on an objective-based TIRF microscope generated by an oil immersion objective (Apo TIRF 100×, 1.49NA, Olympus). Under green laser excitation (532-nm laser, Coherent Inc.), donor and acceptor emissions were split by a dichroic mirror and then imaged on a high-sensitivity electron-multiplying CCD (EMCCD, iXon Ultra 897, Andor). Different acquisition rates were selected regarding the experimentations.

A flow chamber was assembled with PEG surfaces as described[46,47] and was placed on the TIRF microscope. Pol II TECs were immobilized on PEG surface through biotin-streptavidin interactions where biotin can be labeled on Pol II, non-template DNA or RNA separately for different experimental purposes. After washing out free components, a convenient density of fluorescent spots was achieved for single-molecule fluorescence measurements. Fluorescence imaging was

carried out in transcription buffer (20 mM HEPES pH 8.0, 100 mM KCl, 8 mM MgCl$_2$, 0.5 mg ml$^{-1}$ BSA, 7.5 μM ZnCl$_2$, 2 mM DTT) containing oxygen scavenging system: 1 mg ml$^{-1}$ glucose oxidase, 0.4 mg ml$^{-1}$ catalase, 0.8% glucose and 1 mM Trolox (Sigma-Aldrich). Raw fluorescence data ID and IA were extracted from the donor and acceptor channels, respectively, with background corrected.

The TIRF microscope was equipped with alternating-laser excitation (ALEX) module and two lasers (532-nm green laser and 640-nm red laser, Coherent Inc.). ALEX module can control these two lasers to alternatively excite samples and trigger EMCCD imaging simultaneously. This allows alternative imaging of different color fluorophores in real time especially when donor-acceptor distance was beyond the FRET range.

## Single-molecule transcription assay

Pol II TECs were prepared and immobilized in the flow chamber for single-molecule fluorescence imaging. Pol II transcription was restarted by introducing starved nucleotides (ATP, UTP and CTP) with noted concentrations and 2 nM TFIIS (for recovering backtracked Pol II) in transcription buffer with oxygen scavenging system. This will allow Pol II transcription to the first G of G-stretch on non-template DNA which can be monitored via FRET changes for Pol II TECs: bioPol II/RNA-Cy3/DNA-Cy5(+3) and bioPol II/RNA-Cy3/DNA-Cy5(−16). Various RNA lengths can be generated by modifying both the transcription length and/or the RNA length. Transcription termination of Pol II TECs was induced by Sen1 plus starved nucleotides and 2 nM TFIIS. Termination events were monitored via the disappearance of RNA and the fates for other components, such as Sen1, Pol II and DNA, were monitored, respectively, via their specific labels. Duration of fluorescence existence and FRET changes were analyzed via custom matlab codes.

## In vitro transcription assay

This assay was performed as previously described[16]. Briefly, the Pol II TEC for in vitro transcription assay was prepared by sequential assembly of 16-mer-RNA-Cy3(5′), 28nt-Tem, Pol II and 28nt-5′bio-non-Tem components and then tethered to 1 μm-diameter streptavidin-coated superparamagnetic beads (Dynabeads MyOne Streptavidin C1, Life Technologies). After washing away free components, transcription elongation of Pol II TECs was restarted by adding 5 mM ATP, 1 mM each of UTP and CTP. Pol II transcription termination mediated by Sen1 HD was measured by adding 100 nM Sen1 HD, 5 mM ATP, 1 mM UTP, and CTP separately. Reactions were incubated at 28 °C for 15 min and stopped by the addition of 1 μl 0.5 M EDTA and then separated into beads and supernatant fractions. Each fraction was migrated in denatured PAGE (8 M Urea), and the gels were scanned using ChemiDoc MP imaging system (Bio-Rad).

## ATPase activity assay

The ATPase activity of Sen1 HD was measured in 20 μl final volume of transcription buffer containing 10 nM Sen1 HD, 1 mM ATP and 170 ng of various substrates: linear DNA (pUC18 linearized with KpnI), single-stranded DNA (28nt-Tem) and single-stranded RNA (60mer-RNA), respectively. Control experiments were performed by taking out each of the essential components (Sen1 HD, substrate, or ATP). The reactions were carried out at 28 °C for 30 min followed by adding a reagent of an ATPase/GTPase Activity Assay kit (Sigma-Aldrich). The ATP hydrolysis activity was determined by measuring the absorbance at 620 nm using Nanodrop machine.

## Statistics and reproducibility

All data are collected from more than three independent experiments. No statistical method was used to predetermine the sample size. No data were excluded from the analyses. The experiments were not randomized. The Investigators were not blinded to allocation during experiments and outcome assessment. Single-molecule fluorescence

results are analyzed using custom Matlab code (http://github.com/Wang2004w/Code4DataAnalysis/tree/Seg2Analysis4Fluorescence). This code allows manual selection of interested lifetimes from the single-molecule fluorescence data. The extracted duration events are then subjected to kinetic analysis. All fits are performed using Igor Pro (WaveMetrics).

## Single-molecule FRET correction

The presence of PIFE causes an enhancement in the quantum yield of Cy5 dye, which is proportional to the γ factor. As γ can be calculated as $(I_A − I'_A)/(I'_D − I_D)$, where I and I′ are the intensities before and after Cy5 photobleaching[48], the ratio of γ in the presence and absence of Cy5 PIFE equals to the Cy5 PIFE ratio (1.83 for 10 nM Sen1 concentration). Therefore the corrected FRET, named FRET′, is calculated using the following equation[49]:

$$FRET' = \frac{FRET}{\frac{\gamma_{PIFE}}{\gamma_{noPIFE}}(1 - FRET) + FRET} = \frac{FRET}{1.83 \cdot (1 - FRET) + FRET} \quad (1)$$

## Single-molecule Michaelis–Menten analysis

Termination events obtained from bioPol II/RNA-Cy3/DNA-Cy5(+3) TECs (Supplementary Data 1) at 3.5, 5, and 10 nM Sen1 HD concentrations follow a single exponential decay suggesting a rate-limiting step of Sen1 HD binding (Fig. 1c). The histograms are globally fit to a single exponential function:

$$f(\tau) = A \cdot exp(-k_+ S\tau), \quad (2)$$

where $S$ represents Sen1 HD concentration and $k_+$ is the binding rate constant. $S$ values are held during the fit, obtaining $k_+ = (5.5 \pm 0.2) \times 10^6$ M$^{-1}$ s$^{-1}$ with a reduced Chi-square of 1.2 (Supplementary Fig. 4a).

Since Cy3 disappearance which we attributed to RNA release can also be caused by Cy3 photobleaching, the above histograms are then fit with a single exponential function corrected with photobleaching to test the photobleaching effect:

$$f(\tau) = A \cdot exp(-k_+ S\tau) + B \cdot exp\left(-\frac{\tau}{t_0}\right), \quad (3)$$

where $t_0$ is the time constant of Cy3 photobleaching determined in Supplementary Fig. 4a. S values and $t_0 = 237$ s are held during the fit giving $k_+ = (6.7 \pm 0.4) \times 10^6$ M$^{-1}$ s$^{-1}$ with a reduced Chi-square of 1.0 (Supplementary Fig. 4b). The amplitudes reflecting photobleaching effect are $0.009 \pm 0.002$, $0.007 \pm 0.002$, and $0.002 \pm 0.001$ relative to the amplitudes of single exponential fit results: $0.076 \pm 0.008$, $0.123 \pm 0.009$, and $0.117 \pm 0.008$ for 3.5, 5, and 10 nM Sen1 HD concentrations, respectively.

The catalytic rate is determined as $k_{cat} = 4.04$ s$^{-1}$, the inverse of the average duration of Sen1 HD-SNAP649 of 0.25 s (Fig. 2c), using bioPol II/RNA-Cy3/DNA TECs (Supplementary Data 1) and Sen1 HD-SNAP649. A global fit is performed to the above histograms using the single-molecule Michaelis–Menten function[36] corrected with photobleaching:

$$f(\tau) = \frac{k_+^0 k_{cat}}{2a} \{exp[(a+b)\tau] - exp[(b-a)\tau]\} + B \cdot exp\left(-\frac{\tau}{t_0}\right), \quad (4)$$

where

$a = \sqrt{\frac{1}{4}(k_+^0 + k_- + k_{cat})^2 - k_+^0 k_{cat}}$, $b = -\frac{1}{2}(k_+^0 + k_- + k_{cat})$, $k_+^0 = k_+[Sen1HD]$. $k_{cat} = 4.04$ s$^{-1}$ and $t_0 = 237$ s are held during the fit giving $k_+ = (6.8 \pm 13.2) \times 10^6$ M$^{-1}$ s$^{-1}$ and $k_- = (-1.9 \pm 7.8) \times 10^{-16}$ s$^{-1}$ with a reduced Chi-square of 1.0. When setting $k_- = 0$ for the fit, errors are further reduced giving $k_+ = (6.8 \pm 0.4) \times 10^6$ M$^{-1}$ s$^{-1}$ with a reduced Chi-square of

1.0 (Supplementary Fig. 4c). The same fit amplitudes are obtained for photobleaching whereas the amplitudes of single-molecule Michaelis–Menten fit results are $3.18 \pm 0.37$, $3.61 \pm 0.29$, and $1.71 \pm 0.09$ for 3.5, 5, and 10 nM Sen1 HD concentrations, respectively.

The kinetics of Sen1 HD-SNAP649 translocation on RNA to form Sen1 HD-Pol II TEC intermediate is analyzed by fitting the durations of Sen1 HD-SNAP649 action to another version of single-molecule Michaelis–Menten function[36] because of single Sen1 HD-SNAP649 action:

$$f(\tau) = \frac{k_1 k_2}{k_2 - k_1}[\exp(-k_1\tau) - \exp(-k_2\tau)] \qquad (5)$$

where $k_1$ and $k_2$ are fitting parameters. We get $k_1 = 2.20 \pm 0.32$, $2.54 \pm 0.40$, $5.12 \pm 0.98$, and $5.87 \pm 1.19\,\text{s}^{-1}$ (SEM) for 20 μM, 40 μM, 200 μM, and 1.1 mM ATP concentrations, respectively, and $k_2 = 8.69 \pm 2.03\,\text{s}^{-1}$ (Fig. 2c); $k_1 = 0.84 \pm 0.19$, $3.21 \pm 0.68$, $4.59 \pm 1.15$, and $5.60 \pm 1.61\,\text{s}^{-1}$ (SEM) for 20 μM, 40 μM, 200 μM, and 2 mM ATP concentrations, respectively, and $k_2 = 6.71 \pm 2.01\,\text{s}^{-1}$ (Fig. 3b); $k_1 = 4.32 \pm 1.07$, $3.97 \pm 0.86$, $1.34 \pm 0.22$, and $1.46 \pm 0.24\,\text{s}^{-1}$ (SEM) for 14, 28, 58, and 78 nt RNA lengths, respectively, that extend from the RNA exit channel of Pol II and $k_2 = 5.55 \pm 1.48\,\text{s}^{-1}$ (Supplementary Fig. 8a); $k_1 = 4.27 \pm 1.11\,\text{s}^{-1}$ and $k_2 = 8.45 \pm 3.49$, $4.33 \pm 1.31$, $5.88 \pm 2.09$, and $6.56 \pm 2.47\,\text{s}^{-1}$ for Pol II TECs with no mismatch, mismatch$_{+19\sim+28}$, mismatch$_{+19\sim+23}$ and mismatch$_{+24\sim+28}$, respectively (Fig. 5c).

### Reporting summary

Further information on research design is available in the Nature Portfolio Reporting Summary linked to this article.

## Data availability

The data generated in this study are provided in the Source Data and Supplementary Information files and at https://zenodo.org/records/11307425. These single-molecule source data can be read and manipulated by the custom matlab code (http://github.com/Wang2004w/Code4DataAnalysis/tree/Seg2Analysis4Fluorescence). Source data are provided with this paper.

## Code availability

Custom Matlab code used in this paper is available (http://github.com/Wang2004w/Code4DataAnalysis/tree/Seg2Analysis4Fluorescence).

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

## Acknowledgements

This work is supported by the National Natural Science Foundation of China to S. W. (32071228, 12004420), W.H. (12004271, 12274308), M.L. (T2221001, 12090051), C.X. (32171228), W.L. (32270192, 82072240), S.J. (32270037, 32000025), the National Key R&D Program of China to M.L. (2019YFA0709304), the "Strategic Priority Research Program of the Chinese Academy of Sciences to S.W. (XDB37000000), the Youth Innovation Promotion Association of CAS to S.W. (2021009), Y.L. (Y2021003), Key-Area Research and Development Program of Guangdong Province (21202107221900001) and Guangdong Basic and Applied Basic Research Foundation to W.H. (2019A1515110186).

## Author contributions

Y.X., W.H., C.X., J.S., W.L., T.R.S., and S.W. designed research; Y.X., L.W., and S.W. prepared reagents; Y.X., W.H., C.X., J.S., L.W., T.J., Q.J., Y.L., and S.H. performed research; Y.X. and S.W. analyzed the data; Y.X., W.H., C.X., J.S., S.-X.D., W.L., T.R.S., S.W., and M.L. wrote the paper.

## Competing interests

The authors declare no competing interests.
