## [Peer Review File · Nature Communications]

Single-molecule reconstruction of eukaryotic factor-dependent transcription terminationReviewers' Comments:

Reviewer #1:

Remarks to the Author:

In this paper, Xiong et al. employed single-molecule TIRF microscopy to study the composition and dynamics of the yeast Pol II transcription complex before and after transcription termination mediated by Sen1. Using a series of labeling and immobilization strategies, they dissected the kinetic steps in the reaction pathway, upon which they built a model for understanding factor-dependent transcription termination in eukaryotes. The experiments are elegantly designed and performed. Another strength of this manuscript is the quantitative single-molecule data analysis, which enabled the authors to extract mechanistic insights into this complicated and fundamental process.

Overall, this is an excellent study that makes an important contribution to the field. However, I have the following points that the authors should address to further strengthen the paper.

Major points:

- (1) All the termination events in this study were induced in the context of a stalled TEC. Have the authors tried experiments in which Sen1 was added to an actively elongating complex? If not, they should discuss why these experiments are infeasible or unnecessary.
- (2) The PIFE results (e.g., Fig. 1b and S2) need to be better characterized and controlled. Can Cy5 PIFE still be observed without the Cy3 dye? How does the PIFE level correlate with the Pol II position on DNA? It should also be discussed how PIFE might affect the interpretation of the FRET data.
- (3) The conclusion that the hydrolysis of a single ATP by Sen1 triggers termination seems to rely solely on the observation that k_2 is independent on [ATP], but I don't follow the logic. What would be expected if multiple hydrolysis events were required?
- (4) The authors used one rate constant k_1 to describe the ATP-dependent Sen1 translocation, which likely contains multiple kinetic steps and ATP binding/hydrolysis events. This treatment needs to be justified.
- (5) The conclusion that Sen1 continues to translocate on RNA after termination is inferred from the observed dependence of Sen1 residence time on [ATP]. While this is a reasonable speculation, without a more direct assay such as FRET or PIFE, this assertion should be toned down.
- (6) The conclusion that $\sim 1/3$ of Pol II molecules diffuse on DNA post-termination is based on the PIFE signal. The authors examined two Cy5 positions but only showed an example trace for one (Fig. 4f). The results for both should be presented and compared. The authors should also consider using a longer DNA substrate to assess the range of Pol II diffusion and compare it to the E. coli RNAP, which can diffuse for thousands of bps.

Minor points:

- (1) The rationale for choosing a certain function (single exponential, Michaelis-Menten, etc.) for fitting specific types of data needs to be more explicitly stated in the main text.
- (2) Some figures (e.g., Fig. S5) need to include a statistical analysis.
- (3) In Methods, the protein purification and labeling sections should be placed in front of the single-molecule experiment sections.

Reviewer #2:

Remarks to the Author:

In this manuscript, Ying Xiong and colleagues characterize the process of transcription termination mediated by the budding yeast protein Sen1 using single-molecule approaches. This study is a follow-up of a previous work carried out by several of the authors and published in Nat Commun a few years ago. Here, they use a minimal system composed of Pol II elongation complexes and the termination factor Sen1, where several components are fluorescently labeled to monitor their arrival and departure in real time. In brief, they gather data that confirm previous observations: Sen1 binds to the nascent

RNA and utilizes ATP hydrolysis to translocate along the RNA, colliding with Pol II and displacing it from the DNA template. They also generate new quantitative data on parameters such as the speed of Sen1 translocation along the RNA and the time required for Sen1 to dismantle the Pol II elongation complex under their assay conditions.

While the presented data appear to be of high quality, it is important to note that the key steps of the Sen1-dependent termination reaction were previously established by studies from the Libri lab (references 15 and 16 in the manuscript). Additionally, the process was previously characterized at the single-molecule resolution by several authors of the present manuscript (in the aforementioned article in Nat Commun, see reference 17). Therefore, the level of conceptual advance in this study is quite limited. I believe that the data presented here hold some value but are unlikely to be of interest to a broad audience.

Major comments:

- In the introduction, the authors state that "how Sen1 uses ATPs to remodel Pol II TECs and the mechanics of each component (Sen1, Pol II, RNA, and DNA) acting during Sen1-dependent termination remains elusive." I completely agree that it is unclear how the collision of Sen1 with Pol II induces a remodeling of the elongation complex resulting in its dissociation, apart from the established knowledge that ATP is used by Sen1 for translocating on the RNA. However, it does not appear that the authors of this study have addressed this question at all. Regarding the second part of the sentence, the intended meaning of "the mechanics of each component" is unclear to me.

-The authors observe an intermediate of the termination reaction characterized by the co-localization of Sen1, RNA, and Pol II. Given that Sen1 translocation along the nascent RNA to induce termination has been previously demonstrated, the existence of such an intermediate is obvious (in fact, obligatory). Besides providing information about the duration of this intermediate, which is valid for their experimental conditions but may not necessarily hold true in vivo where conditions differ significantly, their results do not offer significant insights into how Sen1 actually dissociates Pol II upon collision, the number of states encompassed within this intermediate (e.g., remodeled elongation complex), etc.

-In this study, the authors characterize the behavior of the helicase domain of Sen1. However, it has been shown that Sen1 interacts with the CTD of Pol II (PMID: 22286094, PMID: 15121901) via a domain adjacent to the helicase domain. Therefore, it is quite likely that the kinetic parameters they determine here for Pol II dissociation are not relevant to the in vivo situation, where Sen1 might be pre-recruited to the elongation complex through recognition of the CTD.

-They observe that, following termination, a fraction of Pol II molecules remain bound to the DNA and diffuse, possibly through nonspecific interactions with the DNA backbone. They suggest that such diffusion might facilitate Pol II recycling/reinitiation at genes downstream of the termination site. However, it seems unlikely that Pol II would diffuse on the chromatin, where DNA is tightly bound to histones. Consequently, the biological relevance of the observation that Pol II seldom diffuses on the DNA remains unclear. Additionally, I am not aware of any studies providing evidence for Pol II recycling on nearby genes after termination. Out of curiosity, I checked the references cited regarding Pol II recycling (35, 40, 42, 43, and 44) and found that 35 and 40 pertain to bacterial RNA pol, representing a very different system, while 42 and 43 are related to Pol III. Although I am not an expert in Pol III transcription, to my knowledge, Pol III recycling to the same gene is not due to Pol III diffusion but rather to the association of Pol III with transcription factors that interact with the entire Pol III transcription unit from the promoter to the terminator. Reference 44 does not report Pol II recycling.

- In the last section of the results, the authors find that rewinding of the transcription bubble is required for the dissociation of Pol II by Sen1. This was already shown in reference 16, using very similar approaches

Reviewer #3:

Remarks to the Author:

This is a really beautiful study that uses single-molecule colocalization to systematically analyze a simplified in vitro transcription termination system for eukaryotic RNA polymerase II. The yeast Sen1 protein functions at many non-coding and cryptic transcript genes, using its ATPase activity to disrupt the elongation complex. While it has been shown to recognize the RNA transcript, it was previously unclear what order the various components dissociated. In this paper, two color TIRF microscopy is used, and each experiment varies how the complex is tethered to the microscope slide and which components are labeled. The results are surprisingly clear, and the interpretations are aided by the ability to see both FRET and PIFE on some of the molecules. Sen1 and RNA depart, leaving a RNA pol II-DNA complex, and then these two subcomplexes eventually dissociate. The final experiment provides interesting data implicating closure of the DNA bubble in polymerase dissociation, although a surprisingly large fraction remains bound for quite some time.

I have some specific (mostly minor) comments, but overall find this a very impressive paper.

1. Last line of page 5: I don't understand what the authors mean when they say that the FRET changes correspond to Pol II initiation. Since this system uses an annealed RNA as primer, no actual initiation occurs. Perhaps they just mean early elongation?
2. Page 7: The abbreviation "ALEX modulation" is first introduced here, but without definition. Only later is it defined as alternating laser excitation.
3. Page 11. It might be worth adding that bubble closure, while necessary for polymerase dissociation, is not sufficient. This point is made clearer in the discussion.
4. Page 13. The "torpedo model" is used in the field to denote RNA pol II termination by the exonuclease Rat1/Xrn2. While I agree that the Sen1 mechanism has some very similar characteristics, I would avoid using the term "torpedo" here to avoid confusion. Later in the paper they point out the similarity to bacterial rho, so "rho-like" would be a better term that others in the field have used to describe Sen1's mechanism.
5. In the abstract and several other places in the paper, the authors state that a single ATP hydrolysis is sufficient for Sen1/RNA dissociation. If you read the paper carefully, it's clear from the ATP titration in the results section that the authors are separating out translocation along the RNA (which hydrolyzes a lot of ATP) from the final dissociation step. But this is not so clear in the abstract and discussion. To avoid confusing the many readers that won't read in depth, I think it's really important that the authors make it clear about the distinction between ATP hydrolysis during translocation versus the final Sen1/RNA dissociation step. Otherwise, it reads as if they are claiming that one ATP is sufficient for both steps.
6. The Data Analysis section of the Methods says custom Matlab codes were used, but no information is given about how they work or how readers could obtain and use them. Is there a way that the tools developed here can be made accessible to others that might be interested? I found that there was a folder for reviewers with the Matlab script, but this wasn't mentioned in the paper. I followed the directions in the ReadMe file, and I got as far as opening the Example trace and picking the time points. However, there was no dialog box, and when I did the final right click the figure did not close. I'm not sure what went wrong, but the authors might ask some other users to try it and see if it works for them.
7. The equations for the Michaelis-Menton fitting are not fully defined or derived. Not all readers will care, but for those that do a section in the Supplemental Info could be really useful for understanding

why there are different versions used for fitting.

8. The rate constants used throughout the paper (k_+ , k_- , k_1 , and k_2) are first shown in Figure 1. But how these correspond to the actual reaction is not shown until Figure 2a. It would be helpful to have that type of diagram right up front instead of just listing the constants underneath.

9. I was surprised to see FRET between the RNA and DNA in the initial elongation complexes, as well as the increase during early elongation (Fig 1b). Is it clear why this is happening, especially late in the reaction when the dye on the RNA should be moving around quite a large volume? The persistence of a low level of both FRET and PIFE has me a little worried that this is actually some bleed through between the red and green channels.

Reviewer #1 (Remarks to the Author):

In this paper, Xiong et al. employed single-molecule TIRF microscopy to study the composition and dynamics of the yeast Pol II transcription complex before and after transcription termination mediated by Sen1. Using a series of labeling and immobilization strategies, they dissected the kinetic steps in the reaction pathway, upon which they built a model for understanding factor-dependent transcription termination in eukaryotes. The experiments are elegantly designed and performed. Another strength of this manuscript is the quantitative single-molecule data analysis, which enabled the authors to extract mechanistic insights into this complicated and fundamental process.

Overall, this is an excellent study that makes an important contribution to the field. However, I have the following points that the authors should address to further strengthen the paper.

Major points:

(1) All the termination events in this study were induced in the context of a stalled TEC. Have the authors tried experiments in which Sen1 was added to an actively elongating complex? If not, they should discuss why these experiments are infeasible or unnecessary.

We thank the reviewer's suggestions and apologize for the missing information. The reason that we used stalled Pol II TECs for all the assays is that Sen1 preferentially terminates paused Pol II TECs rather than elongating TECs as previously reported (NSMB 2013, 20, 884). A statement is added in the main text (line 8 of page 5).

(2) The PIFE results (e.g., Fig. 1b and S2) need to be better characterized and controlled. Can Cy5 PIFE still be observed without the Cy3 dye? How does the PIFE level correlate with the Pol II position on DNA? It should also be discussed how PIFE might affect the interpretation of the FRET data.

We thank the reviewer's suggestions and perform more characterizations below:

Can Cy5 PIFE still be observed without the Cy3 dye? The answer is yes. We prepare Pol II TEC with Cy5 at +3 position of the non-template strand and tether it to the glass surface through biotinylated Pol II. Termination signal is imposed by adding Sen1 and AUC, i.e. dissociation of Cy5 fluorescence from the surface. We observe Cy5 PIFE disappearance before its dissociation, which corresponds to Pol II elongation followed by termination. This result confirms that the existence of Cy5 PIFE is independent of Cy3 dye. A typical trajectory is shown in Supplementary Fig. 2d and a histogram of Cy5 PIFE (characterized as the ratio of the intensity of Cy5 fluorescence with PIFE over that without PIFE) is also presented. Statements are added in line 9 of page 6 in the revised manuscript.

How does the PIFE level correlate with the Pol II position on DNA? To answer this question, we analyze the PIFE value as the ratio of the mean intensity of the Cy5 fluorescence with PIFE over that without PIFE on Pol II TEC with Cy5 at +3 (with various Sen1 concentrations and the backtracking events) and -16 position, respectively. Histograms show that the PIFE ratios fall in the range of 1.66 to 1.83 except 1.50 for the backtracking events with Cy5 at +3 position and 1.98 for Pol II elongation events with Cy5 at +3 position as shown in Fig. 1d and Supplementary Fig. 2. These results suggest a weak correlation of Cy5 PIFE levels with Pol II positions on DNA. Statements are added in lines 9-11 of page 6.

For Cy5 PIFE events during Pol II diffusion on DNA after termination, which is no longer an active transcription

complex but a Pol II molecule remaining bound to DNA, we analyze the PIFE level for Cy5 labeling at the upstream end and downstream end respectively and find ~1.4 and ~1.7 for Cy5 labeling at the downstream and upstream ends respectively (Figure 4h). Statements are added in lines 12-14, and line 23 of page 11.

Discussions on how PIFE may affect FRET data. The presence of Cy5 PIFE during Pol II elongation (grey region) enhances the quantum yield of dye, which is proportional to the γ factor. As γ can be calculated as $(I_A - I'_A)/(I'_D - I_D)$, where I and I' are the intensities before and after Cy5 photobleaching (Nucleic Acids Res. 2009, 37, 5803), the ratio of γ in the presence and absence of Cy5 PIFE equals to the Cy5 PIFE ratio (1.83 for 10 nM Sen1 concentration). Therefore the correct FRET, named FRET', is calculated using the following equation (Sci. rep. 2016, 6, 33257)^{Error! Reference source not found.}:

$$FRET' = \frac{\frac{FRET}{\gamma_{PIFE} - (1-FRET) + FRET}}{\gamma_{noPIFE}} = \frac{FRET}{1.83 \cdot (1-FRET) + FRET}$$

The corrected FRET trajectory is shown in light blue in Figure 1b representing slightly lower FRET values in the grey region. This result tells a more reasonable distance changes between Cy3 and Cy5 dyes during Pol II elongation. This discussion is added in lines 15-17 of page 6 and a subsection "Single-molecule FRET correction" is added in Methods in the revised manuscript.

(3) The conclusion that the hydrolysis of a single ATP by Sen1 triggers termination seems to rely solely on the observation that k_2 is independent on [ATP], but I don't follow the logic. What would be expected if multiple hydrolysis events were required?

We thank the reviewer's comments. The distribution of Sen1 residence time prior to termination displays a fast rise followed by a slow decay indicating that at least two reaction steps are involved. As the slow decay varies with ATP concentrations, the corresponding enzymatic step must depend on ATP concentrations. However, since the rate of the slow decay is not linearly proportional to ATP concentration which can be seen from the histograms of 20 μ M and 1.1 mM in Figure 2c, then an ATP-independent step must exist (Nature 2003, 422, 399). Therefore, we use a two-step model simplified from the single-molecule Michaelis-Menten function to fit and thus separate the Sen1 residence time into two sequential steps: a first step (k_1) dependent on ATP concentration, reflecting multiple cycles of ATP binding and release, and a second step (k_2) which is independent on ATP concentration and would therefore reflect hydrolysis of the single ATP bound by the protein. The ATP-dependent step possibly corresponds to Sen1 translocation along RNA because k_1 also depends on the RNA length. The ATP-independent step possibly corresponds to the final ATP hydrolysis rather than the final ATP binding/hydrolysis cycle prior to Sen1/RNA dissociation because any ATP binding relies on [ATP]. If multiple ATP hydrolysis events are required as the reviewer comments, then multiple ATP binding events will thus be required resulting in a dependence on [ATP]. These multiple binding/hydrolysis events have been separated into the ATP-dependent step as characterized by k_1 . Additionally, a Sen1 molecule has a single ATP binding site (Embo J 2017, 36, 1590) excluding the possibility of multiple ATP binding at a time for the final hydrolysis step.

Corrections are made in the main text (lines 22-27 of page 8 and line 3 of page 9).

(4) The authors used one rate constant k_1 to describe the ATP-dependent Sen1 translocation, which likely contains multiple kinetic steps and ATP binding/hydrolysis events. This treatment needs to be justified.

We thank the reviewer's suggestion. We agree that k_1 describes the multiple steps of ATP binding/hydrolysis events. As described in the response to Q3, we separate the Sen1 actions into two sequential steps: a first step

(k1) dependent on ATP concentration, reflecting multiple cycles of ATP binding and release, and a second step (k2) which is independent of ATP concentration and would therefore reflect hydrolysis of the single ATP bound by the protein. The ATP-dependent step corresponds to Sen1 translocation along RNA, and we use k1 to describe the entire translocation process of Sen1 on RNA. The involved ATP binding/hydrolysis rates are not determined in this study. k2 describes the final ATP hydrolysis step prior to Sen1/RNA dissociation.

Corrections are made in the main text (lines 22-27 of page 8 and line 3 of page 9).

(5) The conclusion that Sen1 continues to translocate on RNA after termination is inferred from the observed dependence of Sen1 residence time on [ATP]. While this is a reasonable speculation, without a more direct assay such as FRET or PIFE, this assertion should be toned down.

We thank and completely agree with the reviewer's suggestions. The dependence of Sen1 residence time on [ATP] post-termination can only suggest ATP binding/hydrolysis of Sen1. Therefore, in the revised manuscript we tone down our conclusion on Sen1 translocation on RNA after termination as the reviewer suggested (line 25 of page 9, and lines 12 and 15 of page 10).

(6) The conclusion that $\sim 1/3$ of Pol II molecules diffuse on DNA post-termination is based on the PIFE signal. The authors examined two Cy5 positions but only showed an example trace for one (Fig. 4f). The results for both should be presented and compared. The authors should also consider using a longer DNA substrate to assess the range of Pol II diffusion and compare it to the *E. coli* RNAP, which can diffuse for thousands of bps.

We thank the reviewer's suggestions. We show the example trace for the condition with Cy5 labeling at DNA ends (Supplementary Fig. 10). We also perform an experiment with longer DNA substrate to characterize the diffusion distance of Pol II post-termination. Firstly, Pol II elongation activity is characterized on this construct by adding 1 mM each of AUGC and 2 nM TFIIS. The observation of Cy5 PIFE suggests Pol II restarting elongation to the downstream DNA end. The elongation time can be estimated as 47 ± 24 s (SD, N = 51) by fitting the duration from injection to the rise of Cy5 PIFE to a single-Gaussian function. This gives a reasonable elongation rate considering a transcription length of (539 + 28) bp (Supplementary Fig. 10). The fraction of Pol II molecules restarting elongation is 51/104 which is comparable to that on bioPol II/RNA-Cy3/DNA539-Cy5(+3) in Figure 1c (328/668). This result confirms Pol II elongation activity as well as that no T4 DNA ligase remaining on DNA which may potentially inhibit Pol II downstream diffusion. In the Sen1-dependent termination experiment using this long DNA construct, no Cy5 PIFE events have been observed in 116 termination events suggesting that Pol II molecules cannot diffuse 539 bp after Sen1-dependent termination, which is a shorter range than that of *E. coli* RNAP diffusion.

To achieve more information on the diffusion activity of Pol II post-termination, we prepare two more Pol II complexes: bioPol II/RNA-Cy3/DNA91-Cy5(5') and bioPol II/RNA-Cy3/DNA49-Cy5(3') to generate two different distances from Cy5 to termination site (91 and 49 bp respectively). The fraction as well as the time for Cy5 PIFE appearance after termination are each plotted versus different distances and fitted to 1D diffusion model (Nat. Commun. 2023, 11: 450) giving fraction (0.31 ± 0.05) and diffusion coefficient (2.4 ± 1.0) $\times 10^{-4}$ $\mu\text{m}^2/\text{s}$ for Pol II after Sen1-dependent termination (Figure 4g).

Additional statements are made in lines 12-23 of page 11, and lines 16-24 of page 18 for Pol II TEC preparation in the revised manuscript. Fitting functions are added in Supplementary Notes.

Minor points:

(1) The rationale for choosing a certain function (single exponential, Michaelis-Menten, etc.) for fitting specific types of data needs to be more explicitly stated in the main text.

We thank the reviewer's suggestions and add statements when using each fit function in the revised manuscript. We also add a section in the Supplementary Notes to describe the rationale for choosing each function for fitting.

Corrections are made in the main text, lines 23-26 of page 6, line 2 of page 7, lines 15-24 of page 8, line 17 of page 9, line 10 of page 10, and line 26 of page 12.

(2) Some figures (e.g., Fig. S5) need to include a statistical analysis.

We thank the reviewer's suggestions. We analyze the Pol II elongation time, the Cy5 PIFE ratio, and the termination time on the Pol II/RNA-Cy3/bioDNA-Cy5(+3) TECs and add these results in Supplementary Fig. 5.

(3) In Methods, the protein purification and labeling sections should be placed in front of the single-molecule experiment sections.

We thank the reviewer's suggestions and move the subsections "Plasmids for protein purification", "Protein purification" and "Sen1 HD-SNAP labeling" in the front of the Methods in the revised text.

Reviewer #2 (Remarks to the Author):

In this manuscript, Ying Xiong and colleagues characterize the process of transcription termination mediated by the budding yeast protein Sen1 using single-molecule approaches. This study is a follow-up of a previous work carried out by several of the authors and published in Nat Commun a few years ago. Here, they use a minimal system composed of Pol II elongation complexes and the termination factor Sen1, where several components are fluorescently labeled to monitor their arrival and departure in real time. In brief, they gather data that confirm previous observations: Sen1 binds to the nascent RNA and utilizes ATP hydrolysis to translocate along the RNA, colliding with Pol II and displacing it from the DNA template. They also generate new quantitative data on parameters such as the speed of Sen1 translocation along the RNA and the time required for Sen1 to dismantle the Pol II elongation complex under their assay conditions.

While the presented data appear to be of high quality, it is important to note that the key steps of the Sen1-dependent termination reaction were previously established by studies from the Libri lab (references 15 and 16 in the manuscript). Additionally, the process was previously characterized at the single-molecule resolution by several authors of the present manuscript (in the aforementioned article in Nat Commun, see reference 17). Therefore, the level of conceptual advance in this study is quite limited. I believe that the data presented here hold some value but are unlikely to be of interest to a broad audience.

Major comments:

- In the introduction, the authors state that "how Sen1 uses ATPs to remodel Pol II TECs and the mechanics of

each component (Sen1, Pol II, RNA, and DNA) acting during Sen1-dependent termination remains elusive." I completely agree that it is unclear how the collision of Sen1 with Pol II induces a remodeling of the elongation complex resulting in its dissociation, apart from the established knowledge that ATP is used by Sen1 for translocating on the RNA. However, it does not appear that the authors of this study have addressed this question at all. Regarding the second part of the sentence, the intended meaning of "the mechanics of each component" is unclear to me.

We thank the reviewer's comments and revise the statement as "how efficiently Sen1 uses ATP hydrolysis to remodel Pol II TECs and, importantly, the fate of each component (Sen1, Pol II, RNA, and DNA) during Sen1-dependent termination remains elusive" (lines 18-20 of page 3).

-The authors observe an intermediate of the termination reaction characterized by the co-localization of Sen1, RNA, and Pol II. Given that Sen1 translocation along the nascent RNA to induce termination has been previously demonstrated, the existence of such an intermediate is obvious (in fact, obligatory). Besides providing information about the duration of this intermediate, which is valid for their experimental conditions but may not necessarily hold true in vivo where conditions differ significantly, their results do not offer significant insights into how Sen1 actually dissociates Pol II upon collision, the number of states encompassed within this intermediate (e.g., remodeled elongation complex), etc.

We thank the reviewer's comments. We agree that the existence of an intermediate can be speculated for Sen1-dependent termination. However, evidence is always necessary and important. In our manuscript, we use single-molecule fluorescence assays to confirm the existence of this intermediate and furthermore, we show the number of each component existed in the intermediate which cannot be speculated. With respect to the question of the number of states encompassed within the Sen1-Pol II intermediate, we provide the new information that a single round of ATP hydrolysis, rather than multiple rounds, is required for dissociation. This implies that a single rate-limiting step corresponding to resolution of a single discrete intermediate state is required to complete termination. It should be noted that there is in fact a nearly infinite number of microscopic substates in any molecular process given atomic vibrations, and that the key is to identify stable states.

-In this study, the authors characterize the behavior of the helicase domain of Sen1. However, it has been shown that Sen1 interacts with the CTD of Pol II (PMID: 22286094, PMID: 15121901) via a domain adjacent to the helicase domain. Therefore, it is quite likely that the kinetic parameters they determine here for Pol II dissociation are not relevant to the in vivo situation, where Sen1 might be pre-recruited to the elongation complex through recognition of the CTD.

We thank the reviewer's comments. We agree that any additional interactions that may exist between Sen1 and Pol II can affect the kinetic parameters derived from our system. However, it has been reported that deletion of the CTD from Pol II doesn't affect the efficiency of Sen1-dependent termination (NSMB 2013, 20, 884) and the authors proposed that the CTD is not required for the dissociation step of Sen1-dependent termination. As a similar experimental system is used in our manuscript but characterized via single-molecule fluorescence assays rather than gel migration, it is reasonable to conclude that the kinetic parameters determined in the manuscript are not dependent on Sen1 interaction with the CTD of Pol II.

-They observe that, following termination, a fraction of Pol II molecules remain bound to the DNA and diffuse, possibly through nonspecific interactions with the DNA backbone. They suggest that such diffusion might facilitate Pol II recycling/reinitiation at genes downstream of the termination site. However, it seems unlikely that Pol II would diffuse on the chromatin, where DNA is tightly bound to histones. Consequently, the biological relevance of the observation that Pol II seldom diffuses on the DNA remains unclear. Additionally, I am not aware of any studies providing evidence for Pol II recycling on nearby genes after termination. Out of curiosity, I checked the references cited regarding Pol II recycling (35, 40, 42, 43, and 44) and found that 35 and 40 pertain to bacterial RNA pol, representing a very different system, while 42 and 43 are related to Pol III. Although I am not an expert in Pol III transcription, to my knowledge, Pol III recycling to the same gene is not due to Pol III diffusion but rather to the association of Pol III with transcription factors that interact with the entire Pol III transcription unit from the promoter to the terminator. Reference 44 does not report Pol II recycling.

We thank the reviewer's comments and completely agree that Pol II diffusion on DNA after termination may be possibly inhibited by nucleosomes *in vivo*. The referee is also correct that references 35 and 40 pertain to bacterial RNA polymerase. Indeed, the observation of diffusing post-termination complexes is novel and emerges from the work on prokaryotic transcription carried out by several laboratories around the world. Thus in this manuscript we provide the first evidence for such a phenomenon in a eukaryotic RNA polymerase, and believe the extension of this intriguing property from the prokaryotic to the eukaryotic world is of high interest and relevance to the field. We have removed references 42 and 43 pertaining to Pol III as well as reference 44, for which we apologize for the mis-citation.

- In the last section of the results, the authors find that rewinding of the transcription bubble is required for the dissociation of Pol II by Sen1. This was already shown in reference 16, using very similar approaches

We thank the reviewer's comments but cannot agree with that. Dissociation of RNA transcript and Pol II are different processes, which occur in discrete steps in time as reported in our manuscript as well as in references 35 and 41. In the first paragraph of the last section of the results, we conclude that the rewinding of transcription bubble is necessary for Pol II dissociation but is insufficient by direct measurements of Pol II dissociation from DNA using single-molecule fluorescence assays. However, reference 16 characterized the dissociation fractions of RNA transcript rather than Pol II possibly with the assumption that Pol II and RNA dissociate simultaneously. We find that only one third of Pol II molecules dissociate simultaneously with RNA dissociation. Therefore reference 16 cannot lead to conclusions on Pol II dissociation.

Additionally, in the second paragraph in the last section of the results, we characterize the dependence of the fractions and kinetics of Sen1/RNA simultaneous dissociation on the rewinding of transcription bubble. These results are completely different from and provide more details than that of reference 16.

Reviewer #3 (Remarks to the Author):

This is a really beautiful study that uses single-molecule colocalization to systematically analyze a simplified in

in vitro transcription termination system for eukaryotic RNA polymerase II. The yeast Sen1 protein functions at many non-coding and cryptic transcript genes, using its ATPase activity to disrupt the elongation complex. While it has been shown to recognize the RNA transcript, it was previously unclear what order the various components dissociated. In this paper, two color TIRF microscopy is used, and each experiment varies how the complex is tethered to the microscope slide and which components are labeled. The results are surprisingly clear, and the interpretations are aided by the ability to see both FRET and PIFE on some of the molecules. Sen1 and RNA depart, leaving a RNA pol II-DNA complex, and then these two subcomplexes eventually dissociate. The final experiment provides interesting data implicating closure of the DNA bubble in polymerase dissociation, although a surprisingly large fraction remains bound for quite some time.

I have some specific (mostly minor) comments, but overall find this a very impressive paper.

1. Last line of page 5: I don't understand what the authors mean when they say that the FRET changes correspond to Pol II initiation. Since this system uses an annealed RNA as primer, no actual initiation occurs. Perhaps they just mean early elongation?

We thank and completely agree with the reviewer's comments. What we meant was the early elongation of Pol II. To avoid misunderstanding of this sentence, we take out the "initiation followed by" as seen in line 1 of page 6.

2. Page 7: The abbreviation "ALEX modulation" is first introduced here, but without definition. Only later is it defined as alternating laser excitation.

We thank the reviewer's suggestions and add the definition as "alternating-laser excitation modulation, ALEX" when first introduced in line 19 of page 5.

3. Page 11. It might be worth adding that bubble closure, while necessary for polymerase dissociation, is not sufficient. This point is made clearer in the discussion.

We thank the reviewer's suggestions and modify our statements in line 9 of page 12 as "rewinding of the transcription bubble is necessary for Pol II dissociation but is insufficient".

4. Page 13. The "torpedo model" is used in the field to denote RNA pol II termination by the exonuclease Rat1/Xrn2. While I agree that the Sen1 mechanism has some very similar characteristics, I would avoid using the term "torpedo" here to avoid confusion. Later in the paper they point out the similarity to bacterial rho, so "rho-like" would be a better term that others in the field have used to describe Sen1's mechanism.

We thank the reviewer's suggestions and replace the term "torpedo" with "rho-like" in the revised text (line 15 of page 3 and line 21 of page 14).

5. In the abstract and several other places in the paper, the authors state that a single ATP hydrolysis is sufficient

for Sen1/RNA dissociation. If you read the paper carefully, it's clear from the ATP titration in the results section that the authors are separating out translocation along the RNA (which hydrolyzes a lot of ATP) from the final dissociation step. But this is not so clear in the abstract and discussion. To avoid confusing the many readers that won't read in depth, I think it's really important that the authors make it clear about the distinction between ATP hydrolysis during translocation versus the final Sen1/RNA dissociation step. Otherwise, it reads as if they are claiming that one ATP is sufficient for both steps.

We thank the reviewer's suggestions. We correct our statements to distinguish the multiple ATP hydrolysis step (translocation) and the single ATP hydrolysis step (dissociation).

In the abstract section (lines 6-8 of page 2), we correct it as "We show that Sen1 takes the RNA transcript as its substrate and translocates along it by hydrolyzing multiple ATPs to form an intermediate with a stalled RNA polymerase II (Pol II) transcription elongation complex (TEC). This intermediate dissociates upon hydrolysis of a single ATP and leads to dissociation of Sen1 and RNA, after which Sen1 remains bound to the RNA."

In the Discussion section (lines 12-14 of page 14), we correct as "Formation of the termination intermediate requires Sen1 translocation along the RNA transcript by hydrolyzing multiple ATPs, while its resolution step requires the hydrolysis of a single ATP molecule, which is sufficient to simultaneously dissociate Sen1 and the RNA transcript".

In line 10 of page 8, we correct "Sen1 translocates on RNA to target Pol II TEC and hydrolyzes a single ATP to dissociate RNA transcript".

6. The Data Analysis section of the Methods says custom Matlab codes were used, but no information is given about how they work or how readers could obtain and use them. Is there a way that the tools developed here can be made accessible to others that might be interested? I found that there was a folder for reviewers with the Matlab script, but this wasn't mentioned in the paper. I followed the directions in the ReadMe file, and I got as far as opening the Example trace and picking the time points. However, there was no dialog box, and when I did the final right click the figure did not close. I'm not sure what went wrong, but the authors might ask some other users to try it and see if it works for them.

We apologize for making a mistake in the readme.txt file that the figure will be automatically closed while the Points.txt file is opened. Although it is complicated to track what was wrong with the code when the reviewer tried the code, we may suggest the newly opened dialog in the (3) step may be closed by any chance. Also we are open for code debugging.

To make a clearer view to use the code, we upload the Matlab code to the GitHub (<http://github.com/Wang2004w/Code4DataAnalysis/tree/Seg2Analysis4Fluorescence>) to make it accessible to people who may be interested and prepare a .docx file with screenshot for each step of the manipulation to track the procedure of using the code. Hope this file can make a clear guide. By the way, the code was developed in Matlab version 2017a and works fine in later versions. We cannot guarantee its working in older versions of Matlab because we haven't tried that.

Statements are added in lines 21-22 of page 20 and Code Availability section.

7. The equations for the Michaelis-Menton fitting are not fully defined or derived. Not all readers will care, but

for those that do a section in the Supplemental Info could be really useful for understanding why there are different versions used for fitting.

We thank the reviewer's suggestions and add a section in the Supplementary Notes. Regarding the suggestions from Reviewer 1, we also add corrections in the main text for better descriptions, lines 23-26 of page 6, line 2 of page 7, lines 15-24 of page 8, line 17 of page 9, line 10 of page 10, and line 26 of page 12.

8. The rate constants used throughout the paper (k_+ , k_- , k_1 , and k_2) are first shown in Figure 1. But how these correspond to the actual reaction is not shown until Figure 2a. It would be helpful to have that type of diagram right up front instead of just listing the constants underneath.

We thank the reviewer's suggestions and modify Figure 1b to show the schematic of the termination reaction.

9. I was surprised to see FRET between the RNA and DNA in the initial elongation complexes, as well as the increase during early elongation (Fig 1b). Is it clear why this is happening, especially late in the reaction when the dye on the RNA should be moving around quite a large volume? The persistence of a low level of both FRET and PIFE has me a little worried that this is actually some bleed through between the red and green channels.

We thank the reviewer's comments and suggestions.

We are also surprised when we first observed the FRET change during the elongation of Pol II TECs (bioPol II/RNA-Cy3/DNA-Cy5(+3)). Together with the FRET result observed on bioPol II/RNA-Cy3/DNA-Cy5(-16), we tend to explain as below:

As the RNA transcript is mainly composed of three nucleotides (A, U, and C) in our experiments, the possibility of forming secondary structures the RNA transcript can be dismissed. Therefore, the behavior of the RNA transcript can be described as a random coil and its persistence length can be estimated as 3 bases from the Freely-Joint Chain (FJC) model for single-stranded nucleic acids. This suggests that the location of the Cy3 at the 5' end of the RNA transcript is roughly floating near the RNA exit channel rather than far away. Although the non-template strand of transcription bubble is highly dynamic, people have obtained the structure of a complete transcription bubble in the presence of TFIIF (Mol. Cell 2015, 59, 258; PDB: 5C4X). In the structure model, we can speculate that Cy5 at +3 position on the non-template strand moves first near to and then far away from the RNA exit channel while Pol II elongates. This could be the origin of the FRET change during Pol II elongation in our results.

As suggested by Reviewer 1, the Cy5 PIFE might affect the FRET value during Pol II elongation state. We try to estimate this effect and obtain slightly reduced FRET values as shown in light blue in Figure 1b. Statements are added in lines 15-17 of page 6.

The leakage of the donor emission into the acceptor channel of our TIRF microscope is 0.08 as previously characterized (Nucleic Acids Res. 2022, 50, 5974). Therefore, the FRET value of ~ 0.2 should be a real FRET in Figure 1b. Another evidence is that the FRET value decreases to about zero when moving Cy5 from +3 position to -16 position which also means that the FRET value (~ 0.2) should be real.

Reviewers' Comments:

Reviewer #1:

Remarks to the Author:

The authors have done a good job addressing my concerns as well as those from the other reviewers. The manuscript is significantly improved and I can recommend its publication. Just one minor comment: the last sentence of Discussion regarding the difference between Pol II and E. coli RNAP's post-termination diffusion could be elaborated a bit more, including the chromatinized DNA in eukaryotes that another reviewer brought up.

Reviewer #2:

Remarks to the Author:

I appreciate the sophisticated in vitro system used in this study and the detailed characterization of kinetic parameters under their reaction conditions. However, I still feel that the level of novelty is limited since the authors mainly confirm previous findings and fail to address the primary gaps in the field. Additionally, while they offer numerous details about the behavior of different molecules in their specific system (albeit with the limitation that they only monitor 1-2 components at a time, inferring the behavior or fate of others), it remains unclear to what extent their findings can be applied or are relevant to the in vivo situation. Nonetheless, it is up to the editor to determine if the level of novelty meets the criteria for publication in this journal.

Other concerns:

1) I find the scheme in Figure 2 to be rather misleading: they depict Pol II bound to the DNA and label it post-termination, yet they haven't presented any data indicating this occurrence since they are not monitoring the presence of Pol II on the DNA (neither the DNA nor Pol II are labeled in these assays). The same applies to Figure 3a; even though they label the DNA here, it is unclear how they conclude from their data that Pol II remains bound to the DNA after termination.

2) In their response to my comments, the authors state that "Dissociation of RNA transcript and Pol II are different processes, which occur in discrete steps in time as reported in our manuscript as well as in references 35 and 41" and that "Therefore reference 16 cannot lead to conclusions on Pol II dissociation".

Firstly, I believe the authors may be referring to a different reference: in 41, there seems to be no evaluation of transcription termination by Sen1 (or any assessment of termination at all).

Secondly, if the authors genuinely believe that the 'Dissociation of RNA transcript and Pol II are different processes' and that their results shed light on the role of transcription bubble rewinding in Pol II dissociation, they should consider changing the final sentence in their results section: "These results suggest that rewinding of the transcription bubble facilitates resolution of the Sen1-Pol II TEC intermediate, i.e. RNA release during Sen1-dependent termination." I also strongly recommend, for clarity and fairness, that they acknowledge in the discussion section the previous data in ref 16, indicating that it was previously shown that rewinding of the transcription bubble increases the efficiency of TEC dissociation by Sen1 (strictly a Pol II molecule bound unspecifically to DNA cannot be considered as a TEC and, thus, RNA release from a TEC is equivalent to TEC dissociation). They should also explain why they believe their findings substantially differ from the earlier ones.

Thirdly, I have significant concerns about their findings and conclusions stating that Pol II often does not dissociate from the DNA simultaneously when the RNA is removed from the TEC. My understanding is that the authors base their conclusions on the observation of PIFE (protein-induced fluorescence enhancement) after transcription termination, which they use as a measure of Pol II

association with the DNA in the proximity of the Cy5 label.

However, I interpret their data differently:

- In fig. 4a: they observe high PIFE when Pol II is near the Cy5 in the DNA (TEC assembly region), once Pol II transcribes until the stalling site the Cy5 signal decreases and remains stable. After termination, Pol II is released in proximity to the end of the DNA template favoring its reassociation with the 3' end either because the template has an overhang (I did not check whether this is the case) or transiently opens (DNA duplex breathing). In the absence of nucleotides Pol II eventually dissociates and the Cy5 signal drops.

- In Fig. 4f, they observe PIFE after RNA release and termination when the Cy5 is positioned at an extremity close to the stalling (and termination) site. This suggests that Pol II release in proximity to the DNA extremity enhances the possibility of Pol II reassociating with the DNA end. Conversely, in Supplementary Fig. 10d, when the Cy5 is positioned farther from the stalling site, they don't observe PIFE. This absence of PIFE might indicate that Pol II is more likely to reassociate with the extremity closer to the termination site.

- When they add NTPs to their transcription reactions using a long template (suppl. fig. 10f), they observe PIFE when Pol II is approaching the end of the template, where Cy5 is located. In this scenario, termination doesn't occur at the original stalling site because no stalling happens upon the addition of the four nucleotides. Termination by Sen1 might occur near the template end if Pol II naturally pauses there. Alternatively, Sen1 might not trigger termination, and RNA release could simply result from Pol II run-off. Post-run-off Pol II might reassociate with the DNA end and potentially engage in aberrant transcription, which is frequently observed in such in vitro systems.

- The fact that Pol II can associate with particular nucleic acid structures is also illustrated by their experiments in Figure 4 where they introduced mismatches to create a static bubble. In these conditions, more than 90% of Pol II molecules remain associated or reassociate with the DNA. Indeed, the static bubble is a structure for which Pol II is particularly avid. This is the principle behind the use of such structures for promoter-independent transcription elongation, a method extensively used before the Kashlev lab developed the approach employed in this study.

- Although not explicitly stated, I presume the authors utilize heparin to support the notion that Pol II mainly stays bound to the template rather than dissociating and reassociating, given the modest effect of heparin addition. However, heparin is expected to outcompete Pol II binding only if Pol II does not have a substantially higher affinity for the structures in the DNA templates than for heparin. Conducting similar experiments using an excess of unlabeled DNA templates instead of heparin could provide more compelling evidence that Pol II does not dissociate and reassociate.

In conclusion, considering their current outcomes, I believe their observations might primarily reflect the non-specific reassociation of Pol II with DNA due to the use of linear DNA as the template for transcription reactions. Consequently, I'm skeptical that these findings shed light on an unforeseen property of Pol II that could enhance our understanding of the termination process in vivo.

3) In page 11 of the revised version: "Pol II elongation activity on this long DNA construct is characterized by adding NTPs (Supplementary Fig. 10e and f). 51 out of 104 Pol II molecules restart transcription elongation consistent with the fraction obtained in Fig. 1c. Levels of Cy5 PIFE when labeling at the downstream (~1.4) and upstream (~1.7) ends of the DNA are presented (Fig. 4h)". This part is not clear to me. Based on the text, I infer that after termination, they introduce NTPs and observe that terminated Pol IIs, either remaining associated or reassociating with the DNA templates, resume elongation. This would be extremely surprising, as extensive research over the years by multiple labs has shown that Pol II alone cannot transcribe from dsDNA (unless nicks or mismatches are present).

However, upon reviewing the figure, as I mentioned in my prior comment (see comment 2 above), my

understanding is that upon adding NTPs at the beginning of the reaction, they observe PIFE at a certain point, which they interpret as evidence of Pol II transcription. I would like to remind that PIFE is not a direct readout of transcription but rather an indication (indirect) of Pol II proximity with the Cy5 label. I recommend the authors offer a more comprehensive explanation of the experiment's rationale, its design, their interpretation of the results, and the crucial insights they believe the experiment offers. At this stage, all this remains unclear to me.

4) Regarding the response to my previous comment about the evidence for the existence of an intermediate Sen1-RNA-Pol II-DNA: strictly in the current study, simultaneous visualization of the four components of this intermediate has not been performed. Instead, they visualize only two components at a time and infer or assume the presence of the other two. While I am strongly convinced about the existence of this complex (as it is the only plausible explanation for previous and present results), I feel the authors might be overstating the case when claiming that the existence of such an intermediate was merely suggested or speculated before, whereas now it is definitively proven.

Reviewer #3:

Remarks to the Author:

The authors have addressed my earlier comments.

REVIEWER COMMENTS

Reviewer #1 (Remarks to the Author):

The authors have done a good job addressing my concerns as well as those from the other reviewers. The manuscript is significantly improved and I can recommend its publication. Just one minor comment: the last sentence of Discussion regarding the difference between Pol II and *E. coli* RNAP's post-termination diffusion could be elaborated a bit more, including the chromatinized DNA in eukaryotes that another reviewer brought up.

We thank the reviewer's suggestion and revise the last sentence of Discussion as "*S. cerevisiae* Pol II possesses a comparable diffusion coefficient but cannot diffuse as far as *E. coli* RNA polymerase Error! Reference source not found., Error! Reference source not found.. This difference might be related to the chromatinized DNA in *S. cerevisiae* but not in *E. coli*, and the biological relevance remains elusive." (page 15 lines 5-8)

Reviewer #2 (Remarks to the Author):

I appreciate the sophisticated in vitro system used in this study and the detailed characterization of kinetic parameters under their reaction conditions. However, I still feel that the level of novelty is limited since the authors mainly confirm previous findings and fail to address the primary gaps in the field. Additionally, while they offer numerous details about the behavior of different molecules in their specific system (albeit with the limitation that they only monitor 1-2 components at a time, inferring the behavior or fate of others), it remains unclear to what extent their findings can be applied or are relevant to the in vivo situation. Nonetheless, it is up to the editor to determine if the level of novelty meets the criteria for publication in this journal.

We disagree with the statement about the novelty of our manuscript, and we are not mainly confirming previous findings. In this manuscript, we answer the long-unknown question whether Sen1 translocates on DNA or RNA to terminate Pol II transcription and this has important consequences for what the nuclear pool of Sen1. Sen1 hydrolyzes a single ATP to terminate the Sen1-Pol II complex, which provides a new view for factor-dependent termination. We also show that, just like *E. coli* Mfd responsible for transcription coupled repair, termination helicases remain associate to their nucleic substrate in post-termination stage. For the first time, we show Pol II fate in post-termination stage. Because of the independence of termination helicases, this Pol II property can shed light on other factor-dependent termination. We provide a kinetic view of Sen1-dependent termination by directly observing the kinetic processes via single-molecule tools, some of which we believe to be common to factor-dependent termination in eukaryotes. These results are relevant to *in vivo* situation.

For the concern that we only monitor one or two components at a time via our single-molecule tools, please read the response to the last concern below (We are observing three out of four components via fluorescence and biotinylation).

Other concerns:

1) I find the scheme in Figure 2 to be rather misleading: they depict Pol II bound to the DNA and label it post-termination, yet they haven't presented any data indicating this occurrence since they are not monitoring the

presence of Pol II on the DNA (neither the DNA nor Pol II are labeled in these assays). The same applies to Figure 3a; even though they label the DNA here, it is unclear how they conclude from their data that Pol II remains bound to the DNA after termination.

We thank the reviewer's suggestion and revise the schemes in Figures 2, 3 and Supplementary Figures 6, 7.

2) In their response to my comments, the authors state that "Dissociation of RNA transcript and Pol II are different processes, which occur in discrete steps in time as reported in our manuscript as well as in references 35 and 41" and that "Therefore reference 16 cannot lead to conclusions on Pol II dissociation".

Firstly, I believe the authors may be referring to a different reference: in 41, there seems to be no evaluation of transcription termination by Sen1 (or any assessment of termination at all).

We apologize for the mis-citation of reference 41, which should be reference 42, in the prior response to this reviewer. The citation was correct in the main text.

Secondly, if the authors genuinely believe that the 'Dissociation of RNA transcript and Pol II are different processes' and that their results shed light on the role of transcription bubble rewinding in Pol II dissociation, they should consider changing the final sentence in their results section: "These results suggest that rewinding of the transcription bubble facilitates resolution of the Sen1-Pol II TEC intermediate, i.e. RNA release during Sen1-dependent termination." I also strongly recommend, for clarity and fairness, that they acknowledge in the discussion section the previous data in ref 16, indicating that it was previously shown that rewinding of the transcription bubble increases the efficiency of TEC dissociation by Sen1 (strictly a Pol II molecule bound unspecifically to DNA cannot be considered as a TEC and, thus, RNA release from a TEC is equivalent to TEC dissociation). They should also explain why they believe their findings substantially differ from the earlier ones. We thank the reviewer's suggestions. We agree with the reviewer's statement in the brackets that Pol II dissociation from TEC is equivalent to RNA release. But the complete dissociation of Pol II from DNA is partially delayed and is slower than that of RNA release, from which we believe that dissociation of RNA transcript and the dissociation of Pol II from DNA are different processes.

The final sentence of the results section is revised as "These results suggest that rewinding of the transcription bubble facilitates RNA release from Pol II TEC during Sen1-dependent termination, yet incomplete dissociation of Pol II from DNA".

We acknowledge reference 16 and explain the difference between this reference and our study in discussion (page 14 lines 24-27).

Thirdly, I have significant concerns about their findings and conclusions stating that Pol II often does not dissociate from the DNA simultaneously when the RNA is removed from the TEC. My understanding is that the authors base their conclusions on the observation of PIFE (protein-induced fluorescence enhancement) after transcription termination, which they use as a measure of Pol II association with the DNA in the proximity of the Cy5 label.

The second sentence of this paragraph is incorrect. Our conclusion (first sentence of this paragraph) is based on the retention of Cy5 fluorescence after RNA release (Cy3 disappearance) as seen from the lowest panel in

figure 4a. This tells that Cy5 labeled DNA remains association to biotinylated Pol II on the glass surface. Cy5 fluorescence will disappear if Pol II separates from the Cy5-DNA. The Cy5 PIFE assay is introduced to localize Pol II which allows detection of Pol II capturing the Cy5 labeled DNA end. The lifetime from RNA release to Cy5 PIFE appearance and its fraction are analyzed supporting a 1D diffusion model for Pol II to move from the termination site to DNA end.

However, I interpret their data differently:

- In fig. 4a: they observe high PIFE when Pol II is near the Cy5 in the DNA (TEC assembly region), once Pol II transcribes until the stalling site the Cy5 signal decreases and remains stable. After termination, Pol II is released in proximity to the end of the DNA template favoring its reassociation with the 3' end either because the template has an overhang (I did not check whether this is the case) or transiently opens (DNA duplex breathing). In the absence of nucleotides Pol II eventually dissociates and the Cy5 signal drops.

We tend to agree with the idea that "After termination, Pol II is released in proximity to the end of the DNA template favoring its reassociation with the 3' end" but don't have evidence to support it. Because being stalled in the presence AUC, Pol II is most probably terminated at this stalling site by Sen1. The terminated Pol II has three fates, one is to dissociate simultaneously when RNA releases as seen from the simultaneous dissociation of RNA and DNA, another is to capture DNA ends as seen from the appearance of Cy5 PIFE, and the other is to remain binding with DNA and not approaching DNA ends as seen from the presence of Cy5 fluorescence but no PIFE. We agree that the short distance between Pol II stalling site and downstream Cy5 labeled DNA end possibly favors Pol II capturing the DNA end.

For the last sentence of this paragraph, nucleotides (AUC) are present through the experiment of figure 4a. Cy5 fluorescence disappearance indicates Pol II dissociation from DNA but in the presence of nucleotides.

- In Fig. 4f, they observe PIFE after RNA release and termination when the Cy5 is positioned at an extremity close to the stalling (and termination) site. This suggests that Pol II release in proximity to the DNA extremity enhances the possibility of Pol II reassociating with the DNA end. Conversely, in Supplementary Fig. 10d, when the Cy5 is positioned farther from the stalling site, they don't observe PIFE. This absence of PIFE might indicate that Pol II is more likely to reassociate with the extremity closer to the termination site.

Agreed. The fraction of Cy5 PIFE decays with the increase in the distance between Pol II stalling (and termination) site and Cy5 DNA end as shown in figure 4g.

- When they add NTPs to their transcription reactions using a long template (suppl. fig. 10f), they observe PIFE when Pol II is approaching the end of the template, where Cy5 is located. In this scenario, termination doesn't occur at the original stalling site because no stalling happens upon the addition of the four nucleotides. Termination by Sen1 might occur near the template end if Pol II naturally pauses there. Alternatively, Sen1 might not trigger termination, and RNA release could simply result from Pol II run-off. Post-run-off Pol II might reassociate with the DNA end and potentially engage in aberrant transcription, which is frequently observed in such in vitro systems.

Supplementary Figure 11g (original supplementary figure 10f) represents the elongation activity of Pol II TECs along the long DNA substrate. In this experiment, freshly prepared Pol II TECs on the long DNA construct restart transcription elongation upon addition of four nucleotides but no Sen1. Thus, RNA release is not caused by

Sen1-dependent termination but by Pol II running off the DNA end. Cy5 PIFE appears when Pol II transcribes to the DNA end. As seen the Cy5 PIFE from Supplementary Figure 11f, the post-run-off Pol II stuck at the DNA end until complete dissociation. Detailed description of this assay is revised (page 11 lines 23-27, page 12 lines 1-4).

- The fact that Pol II can associate with particular nucleic acid structures is also illustrated by their experiments in Figure 4 where they introduced mismatches to create a static bubble. In these conditions, more than 90% of Pol II molecules remain associated or reassociate with the DNA. Indeed, the static bubble is a structure for which Pol II is particularly avid. This is the principle behind the use of such structures for promoter-independent transcription elongation, a method extensively used before the Kashlev lab developed the approach employed in this study.

Agreed.

- Although not explicitly stated, I presume the authors utilize heparin to support the notion that Pol II mainly stays bound to the template rather than dissociating and reassociating, given the modest effect of heparin addition. However, heparin is expected to outcompete Pol II binding only if Pol II does not have a substantially higher affinity for the structures in the DNA templates than for heparin. Conducting similar experiments using an excess of unlabeled DNA templates instead of heparin could provide more compelling evidence that Pol II does not dissociate and reassociate.

Heparin is used to determine whether Pol II remains binding to the opened transcription bubble or nonspecifically to the DNA duplex as has been used for *E. coli* RNA polymerase in reference 42 (page 11 lines 10-11).

We appreciate the reviewer's suggested experiment. However, it possibly cannot provide more compelling evidence as the reviewer expects. Because reference 42 has shown that heparin can compete off nonspecific binding between DNA and *E. coli* RNA polymerase. By replacing heparin with excess of unlabeled DNA which possibly maintains identical binding affinity to that of Cy5 labeled DNA, the nonspecific binding between Pol II and DNA could possibly be competed off as well. Various unlabeled DNA concentrations (100 pM, 1 nM, 10 nM and 100 nM) have been tried and the results are shown in Supplementary Fig. 10, which are consistent with our expectations. The main text is revised (page 11 line 9).

In conclusion, considering their current outcomes, I believe their observations might primarily reflect the non-specific reassociation of Pol II with DNA due to the use of linear DNA as the template for transcription reactions. Consequently, I'm skeptical that these findings shed light on an unforeseen property of Pol II that could enhance our understanding of the termination process in vivo.

We thank the reviewer's careful consideration on Pol II behavior post-termination. We agree that Cy5 PIFE may be possibly caused by the reassociation of Pol II to DNA ends, which is taken as a sign for Pol II approaching DNA ends after termination in the manuscript. As we respond above, Pol II behavior is measured via the retention of Cy5 fluorescence on biotinylated Pol II after termination. The Cy5 PIFE assay is introduced to localize Pol II. The fraction and lifetime for the appearance of Cy5 PIFE are analyzed to determine how Pol II move from the termination site to Cy5 labeled DNA end, which better supports a one-dimensional diffusion

model rather than a three-dimensional diffusion model equivalent to that of the reviewer suggests. A detailed description is listed:

Given that bioPol II/RNA-Cy3/DNA19-Cy5(3') and bioPol II/RNA-Cy3/DNA61-Cy5(5') are prepared with the same DNA construct except that Cy5 is labeled at either the downstream or upstream DNA ends (scheme top 2 and top 4 in Figure 4g), the fraction of Pol II capturing both ends of this DNA construct is 35% (28% + 7%). The fraction of Pol II retention on DNA after termination is 64% in the absence of heparin in Figure 4b. Therefore, about 29% of Pol II molecules (no Cy5 PIFE) remain association with DNA rather than DNA ends after termination, which is inconsistent with the reviewer's interpretation.

To provide more evidence to support our conclusion, we prepare another Pol II TEC with 10 bp between Pol II stalling site and the Cy5 labeled DNA end. The lifetime and the fraction of Cy5 PIFE appearance after termination are analyzed and plotted in the right panel of figure 4g. The data in figure 4g better supports a one-dimensional diffusion model than a three-dimensional diffusion model for Pol II to capture DNA ends. The idea raised by this reviewer that Pol II dissociation and reassociation to DNA ends is three-dimensional diffusion.

The main text is revised (page 11 lines 13, 16-20).

3) In page 11 of the revised version: "Pol II elongation activity on this long DNA construct is characterized by adding NTPs (Supplementary Fig. 10e and f). 51 out of 104 Pol II molecules restart transcription elongation consistent with the fraction obtained in Fig. 1c. Levels of Cy5 PIFE when labeling at the downstream (~1.4) and upstream (~1.7) ends of the DNA are presented (Fig. 4h)". This part is not clear to me. Based on the text, I infer that after termination, they introduce NTPs and observe that terminated Pol IIs, either remaining associated or reassociating with the DNA templates, resume elongation. This would be extremely surprising, as extensive research over the years by multiple labs has shown that Pol II alone cannot transcribe from dsDNA (unless nicks or mismatches are present).

However, upon reviewing the figure, as I mentioned in my prior comment (see comment 2 above), my understanding is that upon adding NTPs at the beginning of the reaction, they observe PIFE at a certain point, which they interpret as evidence of Pol II transcription. I would like to remind that PIFE is not a direct readout of transcription but rather an indication (indirect) of Pol II proximity with the Cy5 label. I recommend the authors offer a more comprehensive explanation of the experiment's rationale, its design, their interpretation of the results, and the crucial insights they believe the experiment offers. At this stage, all this remains unclear to me.

We thank the reviewer's suggestions and revise in the main text "To exclude the possibility that the observation of no Cy5 PIFE is not due to inactive Pol II TECs on this long construct, the elongation activity of the freshly prepared Pol II TECs is measured by adding NTPs. Since Pol II can only reach DNA end via transcription through the 539 bp DNA length, the elongation time can be estimated as the time from NTPs addition until the appearance of Cy5 PIFE reflecting Pol II reaching DNA end, yielding a mean lifetime of 47 ± 24 s (Supplementary Fig. 11f and g). Pol II transcription rate can be estimated as $539 \text{ bp}/47 \text{ s} = 11.5 \text{ bp s}^{-1}$ supporting Pol II elongation activity and is consistent with previous reports ^{Error! Reference source not found., Error! Reference source not found.}. The efficiency of Pol II restarting transcription (51 out of 104 Pol II molecules) is observed consistent with the fraction obtained in Fig. 1c, again supporting active Pol II TECs on the long DNA construct" (page 11 lines 23-27, page 12 lines 1-4). These statements describe how the elongation activity of the freshly prepared Pol II TECs on the long DNA construct is confirmed.

We agree with the reviewer's comments that Cy5 PIFE is not a direct measurement of transcription, but a measurement of Pol II location. Since Pol II can only reach Cy5 position via transcription along DNA in the presence of NTPs, the time from NTPs addition until Cy5 PIFE appearance representing a consistent transcription rate of Pol II ($\sim 11.5 \text{ bp s}^{-1}$) supports Pol II transcription activity.

4) Regarding the response to my previous comment about the evidence for the existence of an intermediate Sen1-RNA-Pol II-DNA: strictly in the current study, simultaneous visualization of the four components of this intermediate has not been performed. Instead, they visualize only two components at a time and infer or assume the presence of the other two. While I am strongly convinced about the existence of this complex (as it is the only plausible explanation for previous and present results), I feel the authors might be overstating the case when claiming that the existence of such an intermediate was merely suggested or speculated before, whereas now it is definitively proven.

It is true that we haven't directly and simultaneously observed all four components, but three out of four components of the Sen1-RNA-Pol II-DNA intermediate in our study. However, our results do prove the existence of all four components in the intermediate.

In the upper part of figure 2a for instance, we prepared Pol II TECs and tether them to streptavidin-coated glass surface via biotinylated Pol II. After washing away untethered components, Cy3 spots were localized on the surface which confirm the existence of both Cy3-RNA and biotinylated Pol II, because Pol II TECs can only be tethered to the surface via biotinylated Pol II. Since Sen1-dependent termination requires sufficiently long RNA transcript, which is longer than the length of Cy3-RNA used for Pol II TEC preparation (please see discussion in page 6 lines 19-20 and reference 15), sufficiently long RNA transcript can only be achieved by Pol II elongation along DNA substrate when subjecting three nucleotides, which confirms the existence of DNA. Finally, Sen1 HD is fluorescently labeled and can be observed directly from the trajectories causing termination.

In the condition of Pol II TEC preparation with biotinylated DNA or RNA in the lower part of figure 2a and figure 3a, Pol II existence was confirmed in the washing step with Ni-NTA magnetic beads (New England Biolabs), which can only bind the his-tag on Pol II. Therefore, any fluorescent spots observed on the surface in the following single-molecule experiments should contain Pol II.

Overall, we believe that our results do prove the existence of all four components of Sen1-RNA-Pol II-DNA intermediate.

For the last sentence, we have cited reference 17 when talking about the intermediate in this study (see page 14 lines 10-11).

Reviewer #3 (Remarks to the Author):

The authors have addressed my earlier comments.

Thanks.

Reviewers' Comments:

Reviewer #2:

Remarks to the Author:

Reviewer #2 (responses to the authors preceded by "R"):

I appreciate the sophisticated in vitro system used in this study and the detailed characterization of kinetic parameters under their reaction conditions. However, I still feel that the level of novelty is limited since the authors mainly confirm previous findings and fail to address the primary gaps in the field. Additionally, while they offer numerous details about the behavior of different molecules in their specific system (albeit with the limitation that they only monitor 1-2 components at a time, inferring the behavior or fate of others), it remains unclear to what extent their findings can be applied or are relevant to the in vivo situation. Nonetheless, it is up to the editor to determine if the level of novelty meets the criteria for publication in this journal.

We disagree with the statement about the novelty of our manuscript, and we are not mainly confirming previous findings. In this manuscript, we answer the long-unknown question whether Sen1 translocates on DNA or RNA to terminate Pol II transcription and this has important consequences for what the nuclear pool of Sen1.

R: this is not correct. The fact that Sen1 translocates along the RNA to terminate Pol II transcription was demonstrated by bulk in vitro transcription termination assays several years ago (PMID: 23748379 and PMID: 28180347) and is consistent with structure-function analyses of Sen1-RNA interaction (PMID: 28408439 and PMID: 37832548) as well as in vivo studies showing the interaction of Sen1 with target transcripts genome-wide (PMID: 32107786). The added value of the present study is the visualization of this process with single-molecule resolution, which was not done before.

Sen1 hydrolyzes a single ATP to terminate the Sen1-Pol II complex, which provides a new view for factor-dependent termination.

R: I must confess that I lack the capacity to appreciate how this finding changes our view of factor-dependent termination.

We also show that, just like E. coli Mfd responsible for transcription coupled repair, termination helicases remain associated to their nucleic substrate in post-termination stage. For the first time, we show Pol II fate in post-termination stage.

R: In early in vitro studies of Sen1-dependent termination about 10 years ago it was shown that Sen1 induces the release of a fraction of Pol IIs from the template DNA that is similar to the fraction of nascent RNA released (PMID: 23748379). In the present study they quantify the process of Pol II release with much higher resolution and find that, at least in their experimental set up, a fraction of polymerases is not released simultaneously with the nascent RNA. This is the new finding of the present study.

Because of the independence of termination helicases, this Pol II property can shed light on other factor-dependent termination. We provide a kinetic view of Sen1-dependent termination by directly observing the kinetic processes via single-molecule tools, some of which we believe to be common to factor-dependent termination in eukaryotes. These results are relevant to in vivo situation.

R: These results might be relevant to the in vivo situation, but to affirm their relevance, experiments such as single-molecule tracking of Pol II and Sen1-dependent transcripts should be conducted in vivo. However, these experiments are very challenging and certainly beyond the scope of this study.

For the concern that we only monitor one or two components at a time via our single-molecule tools, please read the response to the last concern below (We are observing three out of four components via fluorescence and biotinylation).

R: I think there is a misunderstanding here (this response applies to the last point as well). I do believe that even though the authors do not visualize simultaneously the four components of the so-called intermediate (Sen1-RNA-DNA-Pol II complex) in their assays, the results provided in this study

demonstrate the existence of such intermediate. The point I wish to make is that this was already demonstrated a decade ago with the same level of certainty. Indeed, an early study (PMID: 23748379) employed TECs immobilized in streptavidin beads via a biotin in the non-template DNA. Neither the RNA nor Pol II associates with beads in the absence of DNA, so the presence of the DNA is demonstrated by both bridging RNA and Pol II to beads and by supporting transcription by Pol II. The presence of the polymerase is demonstrated by both its DNA-dependent transcription activity and by immunodetection; and the RNA is detected via a fluorescent dye. Because the addition of Sen1 induces the release of both the nascent RNA and Pol II and this requires the interaction of Sen1 with the nascent RNA (degrading or sequestering the RNA prevents Sen1 activity), the only possible explanation for these results is that a Sen1-RNA-DNA-Pol II complex is formed at some point during the termination process. Therefore, I think that the added value of the present study is not the demonstration of the existence of the so-called intermediate but the visualization of several of its components at single-nucleotide resolution and the analysis of the kinetics of formation and dissociation of this complex.

For the rest of my concerns, I find that, in general, the new revisions have clarified the key aspects and significantly improved the manuscript.

REVIEWER COMMENTS

Reviewer #2 (Remarks to the Author):

Reviewer #2 (responses to the authors preceded by “R”):

I appreciate the sophisticated in vitro system used in this study and the detailed characterization of kinetic parameters under their reaction conditions. However, I still feel that the level of novelty is limited since the authors mainly confirm previous findings and fail to address the primary gaps in the field. Additionally, while they offer numerous details about the behavior of different molecules in their specific system (albeit with the limitation that they only monitor 1-2 components at a time, inferring the behavior or fate of others), it remains unclear to what extent their findings can be applied or are relevant to the in vivo situation. Nonetheless, it is up to the editor to determine if the level of novelty meets the criteria for publication in this journal.

We disagree with the statement about the novelty of our manuscript, and we are not mainly confirming previous findings. In this manuscript, we answer the long-unknown question whether Sen1 translocates on DNA or RNA to terminate Pol II transcription and this has important consequences for what the nuclear pool of Sen1.

R: this is not correct. The fact that Sen1 translocates along the RNA to terminate Pol II transcription was demonstrated by bulk in vitro transcription termination assays several years ago (PMID: 23748379 and PMID: 28180347) and is consistent with structure-function analyses of Sen1-RNA interaction (PMID: 28408439 and PMID: 37832548) as well as in vivo studies showing the interaction of Sen1 with target transcripts genome-wide (PMID: 32107786). The added value of the present study is the visualization of this process with single-molecule resolution, which was not done before.

We apologize for being insufficiently accurate and precise in our response to the referee regarding the aspects of our work which are novel. The key novelty of our work is fourfold: 1) we characterize the kinetics of arrival and departure of each of the key components in the reaction, and in particular the arrival of Sen1, the dissociation of RNA, the dissociation of Pol II, the dissociation of DNA, and the dissociation of Sen1; 2) we quantify ATP usage by the helicase; 3) we determine that Sen1 remains associated with RNA after displacement of Pol II; 4) we determine that Pol II can remain associated to DNA after the termination process. We note that we already cite the references mentioned by the reviewer in their appropriate contexts. PMID 23748379 is cited page 9 line 1 as it pertains to the Sen1 translocating on RNA in an ATP-dependent fashion. PMID 28180347 is not a primary source and so is not relevant to cite here. PMID 28408439 and 37832548 are references to structural analysis which are not necessary in the context of our kinetic and compositional analysis. Finally PMID 32107786 refers to genome-wide analysis which lacks mechanistic insight and is also therefore not necessary in the context of our work. To more accurately contextualize our single-molecule results on the observation of Sen1 translocation along RNA as the reviewer mentioned, a revision is made in the abstract (page 2 lines 5, 7, 9 and 10).

Sen1 hydrolyzes a single ATP to terminate the Sen1-Pol II complex, which provides a new view for factor-dependent termination.

R: I must confess that I lack the capacity to appreciate how this finding changes our view of factor-dependent

termination.

The reason we think it may provide a new view for factor-dependent termination is that unless *E. coli* Mfd which hydrolyzes multiple ATPs to translocate along DNA to push stalled RNAP forward until complete dissociation (DOI: 10.1038/nsmb.3019), Sen1 hydrolyzes a single ATP to dissociate Sen1-Pol II complex.

We also show that, just like *E. coli* Mfd responsible for transcription coupled repair, termination helicases remain associated to their nucleic substrate in post-termination stage. For the first time, we show Pol II fate in post-termination stage.

R: In early in vitro studies of Sen1-dependent termination about 10 years ago it was shown that Sen1 induces the release of a fraction of Pol IIs from the template DNA that is similar to the fraction of nascent RNA released (PMID: 23748379). In the present study they quantify the process of Pol II release with much higher resolution and find that, at least in their experimental set up, a fraction of polymerases is not released simultaneously with the nascent RNA. This is the new finding of the present study.

We agree with the reviewer that our detailed characterization of the fate of Pol II upon termination is novel, with a significant fraction of Pol II remaining associated to the DNA post-termination. As pointed out by the reviewer, PMID 23748379 was only able to approximately observe that release of RNA and release of Pol II were in similar amounts, with no real quantification carried out. We also again point out that the new findings of the study include the observation that Sen1 itself remains associated to the RNA post-termination, raising interesting new possibilities for others in the field to explore such as the possibility that Sen1 recycling in vivo could be rate-limiting, or that other helicases may be involved in clearing Sen1 from the RNA. PMID 23748379 is cited when discussing Pol II/RNA dissociation in the revised manuscript (page 11 line 4, page 15 line 4).

Because of the independence of termination helicases, this Pol II property can shed light on other factor-dependent termination. We provide a kinetic view of Sen1-dependent termination by directly observing the kinetic processes via single-molecule tools, some of which we believe to be common to factor-dependent termination in eukaryotes. These results are relevant to in vivo situation.

R: These results might be relevant to the in vivo situation, but to affirm their relevance, experiments such as single-molecule tracking of Pol II and Sen1-dependent transcripts should be conducted in vivo. However, these experiments are very challenging and certainly beyond the scope of this study.

We agree with the reviewer. At the same time, we wish to underscore the fact that these results provide a novel framework within which to consider Sen1 function in vivo, in particular with respect to the mechanism and kinetics for release of Sen1 from the terminated RNA, where it is presumably no longer needed, and which must take place in a timely fashion for Sen1 to be able to work on other targets.

For the concern that we only monitor one or two components at a time via our single-molecule tools, please read the response to the last concern below (We are observing three out of four components via fluorescence and biotinylation).

R: I think there is a misunderstanding here (this response applies to the last point as well). I do believe that even though the authors do not visualize simultaneously the four components of the so-called intermediate (Sen1-RNA-DNA-Pol II complex) in their assays, the results provided in this study demonstrate the existence of such intermediate. The point I wish to make is that this was already demonstrated a decade ago with the same

level of certainty. Indeed, an early study (PMID: 23748379) employed TECs immobilized in streptavidin beads via a biotin in the non-template DNA. Neither the RNA nor Pol II associates with beads in the absence of DNA, so the presence of the DNA is demonstrated by both bridging RNA and Pol II to beads and by supporting transcription by Pol II. The presence of the polymerase is demonstrated by both its DNA-dependent transcription activity and by immunodetection; and the RNA is detected via a fluorescent dye. Because the addition of Sen1 induces the release of both the nascent RNA and Pol II and this requires the interaction of Sen1 with the nascent RNA (degrading or sequestering the RNA prevents Sen1 activity), the only possible explanation for these results is that a Sen1-RNA-DNA-Pol II complex is formed at some point during the termination process. Therefore, I think that the added value of the present study is not the demonstration of the existence of the so-called intermediate but the visualization of several of its components at single-nucleotide resolution and the analysis of the kinetics of formation and dissociation of this complex.

We agree with the reviewer's comments and revise our manuscript to emphasize this aspect of the novelty of our work (page 8 line 12, page 14 lines 11 and 13).

For the rest of my concerns, I find that, in general, the new revisions have clarified the key aspects and significantly improved the manuscript.

Reviewers' Comments:

Reviewer #2:

Remarks to the Author:

The authors have made a reasonable effort to improve the accuracy in contextualizing their results and citing previous studies. The result is satisfactory.

REVIEWERS' COMMENTS

Reviewer #2 (Remarks to the Author):

The authors have made a reasonable effort to improve the accuracy in contextualizing their results and citing previous studies. The result is satisfactory.

We thank all the reviewers for their time and efforts during the peer review process.